



**Research article**
**Title**
Spatially asynchronous changes in strength and stability of terrestrial net ecosystem
productivity
**Running title**
Spatial variability in terrestrial NEP
**Authors**
Erqian Cui[1,2] (eqcui@stu.ecnu.edu.cn)
Chenyu Bian[1,2] (cybian@stu.ecnu.edu.cn)
Yiqi Luo[3] (yiqi.luo@nau.edu)
Shuli Niu[4,5] (sniu@igsnrr.ac.cn)
Yingping Wang[6] (Yingping.Wang@csiro.au)
Jianyang Xia[1,2,*] (jyxia@des.ecnu.edu.cn)
**Affiliations**
[1]Zhejiang Tiantong Forest Ecosystem National Observation and Research Station, Shanghai
Key Lab for Urban Ecological Processes and Eco-Restoration, School of Ecological and
Environmental Sciences, East China Normal University, Shanghai 200241, China;
[2]Research Center for Global Change and Ecological Forecasting, East China Normal
University, Shanghai 200241, China;
[3]Center for ecosystem science and society, Northern Arizona University, Arizona, Flagstaff,
AZ 86011, USA.
[4]Key Laboratory of Ecosystem Network Observation and Modeling, Institute of Geographic
Sciences and Natural Resources Research, Chinese Academy of Sciences, Beijing, China;
[5]University of Chinese Academy of Sciences, Beijing, China;
[6]CSIRO Oceans and Atmosphere, PMB 1, Aspendale, Victoria 3195, Australia.
**Correspondence**
Jianyang Xia, School of Ecological and Environmental Sciences, East China Normal
University, Shanghai 200241, China.
Email: jyxia@des.ecnu.edu.cn
**Key words**
Net ecosystem productivity, spatial asynchronous, $CO_2$ uptake and release, local indicators,
model



**Abstract**

Multiple lines of evidence have demonstrated the persistence of global land carbon (C) sink during the past several decades. However, both annual net ecosystem productivity (NEP) and its inter-annual variation ($IAV_{NEP}$) keep varying over space. Thus, identifying local indicators for the spatially varying NEP and $IAV_{NEP}$ is critical for locating the major and sustainable C sinks on the land. Here, based on a machine-learning-derived database, we first showed that the variations of NEP and $IAV_{NEP}$ are spatially asynchronous. Then, based on daily NEP observations from eddy covariance sites, we found robust logarithmic correlation between annual NEP and ratio of total $CO_2$ exchanges during net uptake ($U$) and release ($R$) periods (i.e., $U/R$). The cross-site variation of mean annual NEP can be linearly indicated by $\ln(U/R)$, while the spatial distribution of $IAV_{NEP}$ was well indicated by the slope (i.e., $\beta$) of the demonstrated logarithmic correlation. Among biomes, for example, forests and croplands had the largest $U/R$ ratio (1.06 ± 0.83) and $\beta$ (473 ± 112 g C m$^{-2}$ yr$^{-1}$), indicating the highest NEP and $IAV_{NEP}$ in forests and croplands, respectively. We further showed that the spatial variations of NEP and $IAV_{NEP}$ were both underestimated by the machine-learning-based and process-based global models. Overall, this study underscores the asynchronously changes in the strength and stability of land C sinks over space, and provides two simple local indicators for their intricate spatial variations. These indicators could be helpful for locating the persistent terrestrial C sinks and provides valuable constraints for improving the simulation of land-atmospheric C exchanges.



## 1. Introduction

Terrestrial ecosystems reabsorb about one-quarter of anthropogenic $CO_2$ emission (Ciais et al., 2019) and are primarily responsible for the recent temporal fluctuations of the measured atmospheric $CO_2$ growth rate (Randerson, 2013; Le Quéré et al., 2018). However, evidence based on eddy-flux measurements (Baldocchi, Chu, & Reichstein, 2018; Rödenbeck, Zaehle, Keeling, & Heimann, 2018), aircraft atmospheric budgets (Peylin et al., 2013), and process-based model simulations (Poulter et al., 2014; Ahlstrom et al., 2015) has shown a large spatial variability in net ecosystem productivity (NEP) on the land. The elusive variation of terrestrial NEP over space refers to both of the dramatic varying mean annual NEP and the divergent inter-annual variability (IAV) in NEP (i.e., $IAV_{NEP}$; usually quantified as the standard deviation of annual NEP) across space (Baldocchi, Chu, & Reichstein, 2018; Marcolla, Rödenbeck, & Cescatti, 2017). The mean annual NEP is related to the strength of carbon sink of a specific ecosystem (Randerson, Chapin III, Harden, Neff, & Harmon, 2002; Luo, & Weng, 2011; Jung et al., 2017), while $IAV_{NEP}$ characterizes the stability of such carbon sink (Musavi et al., 2017). Thus, whether and how NEP and $IAV_{NEP}$ change asynchronously over the space is important for predicting the future locations of carbon sinks on the land (Yu et al., 2014; Niu et al., 2017).

The NEP in terrestrial ecosystems is determined by two components, including vegetation photosynthesis and ecosystem respiration (Reichstein et al., 2005). Because there is a strong covariance between photosynthesis and respiration over space (Baldocchi, Sturtevant, & Contributors, 2015; Biederman et al., 2016), their relative difference could determine the spatial variation of NEP. Many previous analyses have attributed the $IAV_{NEP}$ at the site level to the different sensitivities of ecosystem photosynthesis and respiration to environmental fluctuations among years (Gilmanov et al., 2005; Reichstein et al., 2005; Musavi, 2017). For example, some studies have reported that $IAV_{NEP}$ is more associated with variations in photosynthesis than carbon release (Ahlstrom et al., 2015; Novick, Oishi, Ward, Siqueira, Juang, & Stoy, 2015; Li et al., 2017), whereas others have indicated that respiration is more sensitive to anomalous climate variability (Valentini et al., 2000; von Buttlar et al., 2017). Alternatively, the annual NEP of a given ecosystem can be defined numerically as the balance



between the $CO_2$ uptake and release processes (Gray et al., 2014), which are more direct
components for NEP (Fu et al., 2019). It is still unclear whether ecosystem $CO_2$ uptake and
release could be integrated into some simple indicators for the spatially varying NEP and
$IAV_{NEP}$ in terrestrial ecosystems.

Conceptually, the total $CO_2$ uptake flux ($U$) is determined by the length of $CO_2$ uptake

period ($CUP$) and the $CO_2$ uptake rate, while the total $CO_2$ release flux ($R$) depends on the
length of $CO_2$ release period ($CRP$) and the $CO_2$ release rate (Fig. 2b). The variations of NEP
thus should be innovatively attributed to these decomposed components. A strong spatial
correlation between mean annual NEP and length of $CO_2$ uptake period has been reported in
evergreen needle- and broad-leaved forests (Churkina, Schimel, Braswell, & Xiao, 2005;
Richardson, Keenan, Migliavacca, Ryu, Sonnentag, & Toomey, 2013; Keenan et al., 2014),
whereas atmospheric inversion data and vegetation photosynthesis model indicated a dominant
role of the maximal carbon uptake rate (Fu, Dong, Zhou, Stoy, & Niu, 2017; Zhou et al., 2017).
However, the relative importance of these phenological and physiological indicators for the
spatially varying NEP remains unclear.

In this study, we first explored the changes in NEP and $IAV_{NEP}$ at the global scale based

on data from a widely-used machine-learning-derived product (i.e., FLUXCOM). To address
the local indicators for spatially varying NEP, we decomposed annual NEP into $U$ and $R$. Then,
we examined the relationship of $NEP \propto \frac{U}{R}$ based on the observations at 72 eddy covariance
towers which has >5 years measurements in the FLUXNET2015 Dataset (Jung et al., 2017). In
addition, we used the observations to evaluate the spatial variations of NEP and $IAV_{NEP}$ in the
FLUXCOM database and a process-based model (CLM4.5) (Oleson et al., 2013). The major
aim of this study is to explore whether there are useful local indicators for the spatially varying
NEP and $IAV_{NEP}$ in terrestrial ecosystems.
**2. Materials and Methods**
**2.1 Datasets**
Daily NEP observations of eddy covariance sites were obtained from the FLUXNET2015 Tier
1 dataset (http://fluxnet.fluxdata.org/data/fluxnet2015-dataset/). The FLUXNET2015 dataset



provides half-hourly data of carbon, water and energy fluxes at over 210 sites that are
standardized and gap-filled (Pastorello et al., 2017). However, time series of most sites are still
too short for the analysis of inter-annual variation in NEP. So only the sites that provided the
availability of eddy covariance flux measurements for at least 5 years are selected. This leads
to a global dataset of 72 sites with different biomes across different climatic regions. Based on
the biome classification from the International Geosphere-Biosphere Programme (IGBP)
provided for the FLUXNET2015 sites, the selected sites include 35 forests (FOR), 15
grasslands (GRA), 11 croplands (CRO), 4 wetlands (WET), 2 shrublands (SHR) and 5
savannas (SAV) (Fig. S1 and Table S1). The stand age information of forest sites is the average
tree age of the stand, and was obtained from the Biological Ancillary Disturbance and
Metadata (BAMD) of the FLUXNET dataset (Musavi, et al., 2017).

The FLUXCOM dataset presents an upscaling of carbon flux estimates from 224 flux

tower sites based on multiple machine learning algorithms and satellite data (Jung et al., 2017).
Meteorological measurements from CRUNCEPv6 and a serious of remotely sensed datasets
were used as input. For this study, we downloaded the NEP product from the Data Portal of the
Max Planck Institute for Biochemistry (https://www.bgc-jena.mpg.de). Daily outputs from
FLUXCOM for the period 1980-2013 were used to map the spatial variation in terrestrial NEP
and calculate the local indicators for the spatially varying NEP at the same locations of the flux
tower sites.

Daily NEP simulations from Community Land Model version 4.5 (CLM4.5) were also

used to calculate the local indicators for the spatially varying NEP at the corresponding flux
tower sites. We ran the CLM4.5 model from 1990 to 2010 with a spatial resolution of 1° to
match the available FLUXCOM dataset. Here, NEP was derived as the difference between
GPP and TER, and TER was calculated as the sum of simulated autotrophic and heterotrophic
respiration. The daily outputs from CLM4.5 were used to calculate the local indicators for the
spatially varying NEP at the same locations of the flux tower sites.
**2.2 Decomposition of NEP and the calculations for its local indicators**



The annual NEP of a given ecosystem can be defined numerically as the difference between
the $CO_2$ uptake and release. As illustrated in Figure 2b:

$$NEP = U - R \tag{1}$$

where the total $CO_2$ uptake flux ($U$) and the total $CO_2$ release flux ($R$) can be further
decomposed as:

$$U = \bar{U} \times CUP \tag{2}$$

$$R = \bar{R} \times CRP \tag{3}$$

where the $\bar{U}$ (g C m$^{-2}$ d$^{-1}$) is the mean daily $CO_2$ uptake over $CUP$ and $\bar{R}$ (g C m$^{-2}$ d$^{-1}$)
represents the mean daily $CO_2$ release over $CRP$. The calculations of these direct indicators are
as follows:

$$U = \sum_{i=1}^{m} NEP_i \ (NEP_i > 0; \ CUP = m) \tag{4}$$

$$R = \sum_{i=1}^{n} NEP_i \ (NEP_i < 0; \ CRP = n) \tag{5}$$

where $NEP_i$ refers to the daily NEP (g C m$^{-2}$ d$^{-1}$) in the $ith$ day. Because many studies have
reported that the vegetation $CO_2$ uptake during the growing season and the non-growing soil
respiration are tightly correlated (Luo, & Zhou, 2006; Xia, Chen, Piao, Ciais, Luo, & Wan,
2014; Zhao, Peichl, Öquist, & Nilsson, 2016), we further tested the relationship between
annual NEP and the ratio of $\frac{U}{R}$ (i.e., $NEP \propto \frac{U}{R}$). Then we found that annual NEP was closely
related with the ratio of $\frac{U}{R}$ (Figure S2). Therefore, NEP in any year of any given ecosystem
can be expressed as:

$$NEP = \beta \cdot \ln\left(\frac{U}{R}\right) \tag{6}$$

where the parameter $\beta$ represents the slope of the linear relationship of $NEP \propto \ln\left(\frac{U}{R}\right)$. Based
on the definitions of $U$ and $R$, the ratio $\frac{U}{R}$ can be further written as:

$$\frac{U}{R} = \frac{\bar{U}}{\bar{R}} \cdot \frac{CUP}{CRP} \tag{7}$$





These components of NEP contain both photosynthesis and respiration flux, which
directly indicate the net $CO_2$ exchange of an ecosystem. Ecologically, the ratio of $\frac{\bar{U}}{\bar{R}}$ reflects
the relative physiological difference between ecosystem $CO_2$ uptake and release strength,
while the ratio of $\frac{CUP}{CRP}$ is an indicator of net ecosystem $CO_2$ exchange phenology.
Environmental changes may regulate these ecological processes and ultimately affect the
ecosystem NEP. The slope $\beta$ indicates the response sensitivity of NEP to the changes in
phenology and physiological processes. All of $\beta$, $\frac{CUP}{CRP}$ and $\frac{\bar{U}}{\bar{R}}$ were then calculated from the
selected eddy covariance sites and the corresponding pixels of these sites in models. These
derived indicators from eddy covariance sites were then used to benchmark the results
extracted from the same locations in models.
**2.4 Calculation of the relative contributions**
To further identify the relative contributions of $\frac{\bar{U}}{\bar{R}}$ and $\frac{CUP}{CRP}$ in driving the spatiotemporal
variations in the local indicator $\frac{U}{R}$, we linearized the equation (7) as
$$\log\left(\frac{U}{R}\right) = \log\left(\frac{\bar{U}}{\bar{R}}\right) + \log\left(\frac{CUP}{CRP}\right) \qquad (8)$$
Then we used a relative importance analysis method to quantify the relative contributions
of each ratio to the spatiotemporal variations in $\frac{U}{R}$. The algorithm was performed with the
"ralaimpo" package in R (R Development Core Team, 2011). The "relaimpo" package is based
on variance decomposition for multiple linear regression models. We chose the most
commonly used method named "Lindeman-Merenda-Gold (LMG)" (Grömping, 2007) from
the methods provided by the "ralaimpo" package. This method allows us to quantify the
contributions of explanatory variables in a multiple linear regression model. In each site, we
calculated the contributions of $\frac{\bar{U}}{\bar{R}}$ and $\frac{CUP}{CRP}$ in explaining inter-annual variation in $\frac{U}{R}$. Across
the 72 FLUXNET sites, we quantified the relative importance of $\frac{\bar{U}}{\bar{R}}$ and $\frac{CUP}{CRP}$ to cross-site
changes in $\frac{U}{R}$.
**3. Results**





### 3.1 Spatial variability in terrestrial NEP

Based on the FLUXCOM product, a large spatial variation in terrestrial NEP and $IAV_{NEP}$ existed over 1980-2013. The tropical forests were typically large carbon sinks accompanied by considerable interannual variability. On the contrary, the boreal tundra ecosystems were stable carbon sinks and the shrublands in the Southern Hemisphere were variable carbon sources (Fig. 1a). This remarkable spatial difference in terrestrial NEP was particularly obvious from eddy-flux measurements (Fig. S1), and the global average IAV of NEP ($175 \pm 111$ g C m$^{-2}$ yr$^{-1}$) was large relative to global annual mean NEP ($216 \pm 234$ g C m$^{-2}$ yr$^{-1}$). These spatial patterns were also supported by the model outputs (Jung et al., 2017) and atmospheric inversion product (Marcolla, Rödenbeck, & Cescatti, 2017).

More importantly, we found that the variations of NEP and $IAV_{NEP}$ were spatially asynchronous. Along the latitudinal gradients, terrestrial NEP peaked at equatorial regions, whereas the highest $IAV_{NEP}$ existed in semiarid regions near 37º S (Fig. 1b). The demonstrated spatial asynchrony further revealed the necessary to identify local indicators for the spatially varying NEP and $IAV_{NEP}$, separately.

### 3.2 Local indicators for spatially varying NEP

To find local indicators for the spatially varying NEP in terrestrial ecosystems, we first tested the relationship between NEP and the $\frac{U}{R}$ ratio across the 72 flux-tower sites. We found robust logarithmic correlation between annual NEP and $\frac{U}{R}$ at all sites (Fig. 2a; Fig. S2), with ~90% of $R^2$ falling within a range from 0.7 to 1 (Fig. 2c). Across the 72 flux-tower sites, the spatial changes in mean annual NEP were significantly correlated to $\ln\left(\frac{U}{R}\right)$ ($R^2 = 0.65$, $P < 0.01$) (Fig. 3a). This finding suggests that the mean annual ratio $\ln\left(\frac{U}{R}\right)$ is a good indicator for NEP and its spatial variation. By contrast, the spatial variation of $IAV_{NEP}$ was well explained by the slope (i.e., $\beta$) of the temporal correlation between NEP and $\ln\left(\frac{U}{R}\right)$ at each site ($R^2 = 0.39$, $P < 0.01$; Fig. 3b) rather than $\ln\left(\frac{U}{R}\right)$ (Fig. S3). The wide range of ratio $\beta$ reveals a large divergence of NEP sensitivity across biomes, ranging from $121 \pm 118$ g C m$^{-2}$ yr$^{-1}$ in shrubland to $473 \pm 112$ g C m$^{-2}$ yr$^{-1}$ in cropland.



The decomposition of indicator $\frac{U}{R}$ into $\frac{\bar{U}}{\bar{R}}$ and $\frac{CUP}{CRP}$ allowed us to quantify the relative
importance of these two ratios in driving $\frac{U}{R}$ variability. The linear regression and relative
importance analysis showed a more important role of $\frac{CUP}{CRP}$ (81%) than $\frac{\bar{U}}{\bar{R}}$ (19%) in explaining
the cross-site variation of $\frac{U}{R}$ (Fig. 4). Therefore, the spatial distribution of mean annual NEP
was mostly driven by the phenological rather than physiological changes.

**3.3 Simulated spatial variations in NEP by models**

We further used these two simple indicators (i.e., $\frac{U}{R}$ and $\beta$) to evaluate the simulated spatial
variations of NEP by the machine-learning approach (i.e., FLUXCOM) and a widely-used
process-based model (i.e., CLM4.5). We found that both of FLUXCOM and CLM4.5
underestimated the spatial variation of mean annual NEP and $IAV_{NEP}$ (Fig. 5a). The low spatial
variation of mean annual NEP in FLUXCOM and CLM4.5 could be inferred from their more
converging $\ln\left(\frac{U}{R}\right)$ than flux-tower measurements (Fig. 5b). The underestimated variation of
$IAV_{NEP}$ in these modeling results was also clearly shown by the smaller $\beta$ values (268.22,
126.00 and 145.08 for FLUXNET, FLUXCOM and CLM4.5, respectively) (Fig. 5b).

**4. Discussion**

**4.1 New perspective for locating the major and sustainable land C sinks**

Large spatial differences of mean annual NEP and $IAV_{NEP}$ have been well-documented in
previous studies (Jung et al., 2017; Marcolla, Rödenbeck, & Cescatti, 2017; Fu et al., 2019).
Here we provide a new perspective for quantifying the spatially varying NEP by tracing
annual NEP into several local indicators. Therefore, these traceable indicators could provide
useful constraints for predicting annual NEP, especially in areas without eddy-covariance
towers.
Typically, the C sink capacity and its stability of a specific ecosystem are characterized
separately (Keenan et al., 2014; Ahlstrom et al., 2015; Jung et al., 2017). Here we integrated
NEP into two simple indicators that could directly locate the major and sustainable land C sink.
Among biomes, forests and croplands had the largest $\ln\left(\frac{U}{R}\right)$ and $\beta$, indicating the strongest





and the most unstable C sink in forests and croplands, respectively. The highest $\beta$ in croplands
implies that the rapid global expansion of cropland may enlarge the $IAV_{NEP}$ on the land. In fact,
the cropland expansion has been confirmed as one important driver of the recent increasing
global vegetation growth peak (Huang et al., 2018) and atmospheric $CO_2$ seasonal amplitude
(Gary et al., 2014; Zeng et al., 2014).

**4.2 Phenology-dominant spatial distribution of mean annual NEP**

Recent studies have demonstrated that the spatiotemporal variations in terrestrial gross
primary productivity are jointly controlled by plant phenology and physiology (Xia et al., 2015;
Zhou et al., 2016). Here we demonstrated the dominant role of the phenology indicator $\frac{CUP}{CRP}$ in
driving the spatial difference of $\frac{U}{R}$ and therefore the mean annual NEP. The reported low
correlation between $\frac{U}{R}$ and the physiological indicator $\frac{\bar{U}}{\bar{R}}$ could partly be attributed to the
convergence of $\frac{\bar{U}}{\bar{R}}$ across FLUXNET sites (Fig. S4). The convergent $\frac{\bar{U}}{\bar{R}}$ across sites was first
discovered by Churkina *et al.* (2005) as 2.73 ± 1.08 across 28 sites, which included DBF, EBF
and crop/grass. In this study, we found the $\frac{\bar{U}}{\bar{R}}$ across the 72 sites is 2.71 ± 1.61, which
validates the discovery by Churkina *et al*. However, the $\frac{\bar{U}}{\bar{R}}$ varied among biomes (2.86 ± 1.56
for forest, 2.16 ± 1.14 for grassland, 3.47 ± 1.98 for cropland, 2.89 ± 1.47 for wetland, 1.89 ±
1.10 for shrub, 1.83 ± 0.88 for savanna). This spatial convergence of $\frac{\bar{U}}{\bar{R}}$ at the ecosystem level
provides important constraints for global models that simulate various physiological processes
(Peng et al., 2015; Xia et al., 2017). These findings imply that the phenology changes will
greatly affect the locations of the terrestrial carbon sink by modifying the length of carbon
uptake period (Richardson, Keenan, Migliavacca, Ryu, Sonnentag, & Toomey, 2013; Keenan
et al., 2014).

**4.3 The underestimated spatial variations of NEP in models**

This study showed that the considerable spatial variations in mean annual NEP and $IAV_{NEP}$
were both underestimated by the machine-learning-based and process-based global models,
which could also be inferred from their local indicators. The low variations of $\frac{U}{R}$ ratio in the





two modeling approaches could be largely due to their simple representations of the diverse
terrestrial plant communities into a few plant functional types with parameterized properties
(Sakschewski et al., 2015). The ignorance of year-to-year vegetation dynamic could lead to the
smaller $\beta$ by allowing for only limited variations of phenological and physiological responses
to environmental changes (Reichstein, Bahn, Mahecha, Kattge, & Baldocchi, 2014; Kunstler
et al., 2016). Although the magnitude of $IAV_{NEP}$ depends on the spatial resolution (Marcolla,
Rödenbeck, & Cescatti, 2017), we recommend future model benchmarking analyses to use not
only the machine-learning-based data product (Bonan et al., 2018) but also the site-level
measurements or indicators (i.e., $\ln\left(\frac{U}{R}\right)$ and $\beta$).

**4.4 Conclusions and further implications**

In summary, the findings in this study have some important implications for understanding the
variation of NEP on the land. First, forest ecosystems have the largest annual NEP due to the
largest $\ln\left(\frac{U}{R}\right)$ while croplands show the highest $IAV_{NEP}$ because of the highest $\beta$. Second, the
spatial convergence of $\frac{\bar{U}}{\bar{R}}$ suggests a tight linkage between plant growth and the non-growing
season soil microbial activities (Xia, Chen, Piao, Ciais, Luo, & Wan, 2014; Zhao, Peichl,
Öquist, & Nilsson, 2016). However, it remains unclear whether the inter-biome variation in $\frac{U}{R}$
is due to different plant-microbe interactions between biomes. Third, the within-site
convergent but spatially varying $\beta$ needs better understanding. Previous studies have shown
that a rising standard deviation of ecosystem functions could indicate an impending ecological
state transition (Carpenter, & Brock, 2006; Scheffer et al., 2009). Thus, a sudden shift of the
$\beta$-value may be an important early-warning signal for the critical transition of $IAV_{NEP}$ of an
ecosystem.
In addition, considering the limited eddy-covariance sites with long-term observations,
these findings need further validation once the longer time-series of measurements from more
sites and vegetation types become available. Overall, this study highlights the asynchronous
changes in NEP and $IAV_{NEP}$ over space on the land, and provides the $\frac{U}{R}$ ratio and $\beta$ as two
simple local indicators for their spatial variations. These indicators could be helpful for



locating the persistent terrestrial C sinks in where the $\ln\left(\frac{U}{R}\right)$ ratio is high but the $\beta$ is low.
Their estimates based on observations are also valuable for benchmarking and improving the
simulation of land-atmospheric C exchanges in Earth system models.

**295 Acknowledgements**

This work was financially supported by the National Key R&D Program of China
(2017YFA0604600), National Natural Science Foundation of China (31722009, 41630528)
and National 1000 Young Talents Program of China. This work used eddy covariance dataset
acquired and shared by the FLUXNET community, including these networks: AmeriFlux,
AfriFlux, AsiaFlux, CarboAfrica, CarboEuropeIP, CarboItaly, CarboMont, ChinaFlux,
Fluxnet-Canada, GreenGrass, ICOS, KoFlux, LBA, NECC, OzFlux-TERN, TCOS-Siberia,
and USCCC. The ERA-Interim reanalysis data are provided by ECMWF and processed by
LSCE. The FLUXNET eddy covariance data processing and harmonization was carried out by
the European Fluxes Database Cluster, AmeriFlux Management Project, and Fluxdata project
of FLUXNET, with the support of CDIAC and ICOS Ecosystem Thematic Center, and the
OzFlux, ChinaFlux and AsiaFlux offices.
*Data availability statement.* Eddy flux data are available at
http://fluxnet.fluxdata.org/data/fluxnet2015-dataset/; the data supporting the findings of this
study are available within the article and the Supplementary Information.
*Author contribution.* E. Cui and J. Xia devised and conducted the analysis. Y. Luo, S. Niu, Y.
Wang and C. Bian provided critical feedback on the method and results. All authors
contributed to discussion of results and writing the paper.
*Competing interests.* The authors declare that there is no conflict of interest.





**FIGURES**
**Figure 1** Locations of carbon sinks (mean annual NEP) and their stability (IAV$_{NEP}$) on the land.
a, Spatial patterns of mean annual NEP and IAV$_{NEP}$. b, Latitudinal patterns of mean annual
NEP and IAV$_{NEP}$.
**Figure 2** Relationship between annual NEP and $\frac{U}{R}$ for 72 FLUXNET sites (of the form
NEP $= \beta \cdot \ln\left(\frac{U}{R}\right)$). a, Dependence of annual NEP on the ratio between total $CO_2$ exchanges
during net uptake ($U$) and release ($R$) periods (i.e., $\frac{U}{R}$). Each line represents one flux site with
at least 5 years of observations. b, Conceptual figure for the decomposition framework
introduced in this study. Annual NEP can be quantitatively decomposed into the following
indicators: $NEP = U - R$. c, Distribution of the explanation of $\frac{U}{R}$ on temporal variability of
NEP ($R^2$) for FLUXNET sites.
**Figure 3** Contributions of the two indicators in explaining the spatial patterns of mean annual
NEP and IAV$_{NEP}$. a, The relationship between annual mean NEP and $\ln\left(\frac{U}{R}\right)$ across
FLUXNET sites ($R^2 = 0.65$, $P < 0.01$). The insets show the variation of $\ln\left(\frac{U}{R}\right)$ for different
terrestrial biomes. b, The explanation of $\beta$ on IAV$_{NEP}$ ($R^2 = 0.39$, $P < 0.01$). The insets show
the distribution of parameter $\beta$ for different terrestrial biomes. The number of site-years at
each site is indicated with the size of the point.
**Figure 4** The linear regression between $\frac{U}{R}$ with $\frac{CUP}{CRP}$ ($R^2 = 0.71$, $P < 0.01$) and $\frac{\bar{U}}{\bar{R}}$ ($R^2 = 0.09$,
$P < 0.01$) across sites. The insets show the relative contributions of each indicator to the
spatial variation of $\frac{U}{R}$. The number of site-years at each site is indicated with the size of the
point.
**Figure 5** Representations of the spatially varying NEP and its local indicators in FLUXCOM
product and the Community Land Model (CLM4.5). a, The variation of mean annual NEP and
IAV$_{NEP}$ derives from FLUXNET, FLUXCOM and CLM4.5. Variation in mean annual NEP:
the standard deviation of mean annual NEP across sites; Variation in IAV$_{NEP}$: the standard
deviation of IAV$_{NEP}$ across sites. b, Representations of the local indicators for NEP in
FLUXNET, FLUXCOM and CLM4.5. The corresponding distributions of $\ln\left(\frac{U}{R}\right)$ and $\beta$ are



shown at the top and right. Significance of the relationship between annual NEP and $\ln\left(\frac{U}{R}\right)$
for each site is indicated by the circle: closed circles: $P{<}0.05$; open circles: $P{>}0.05$. Note that
the modeled results are from the pixels extracted from the same locations of the flux tower
sites.





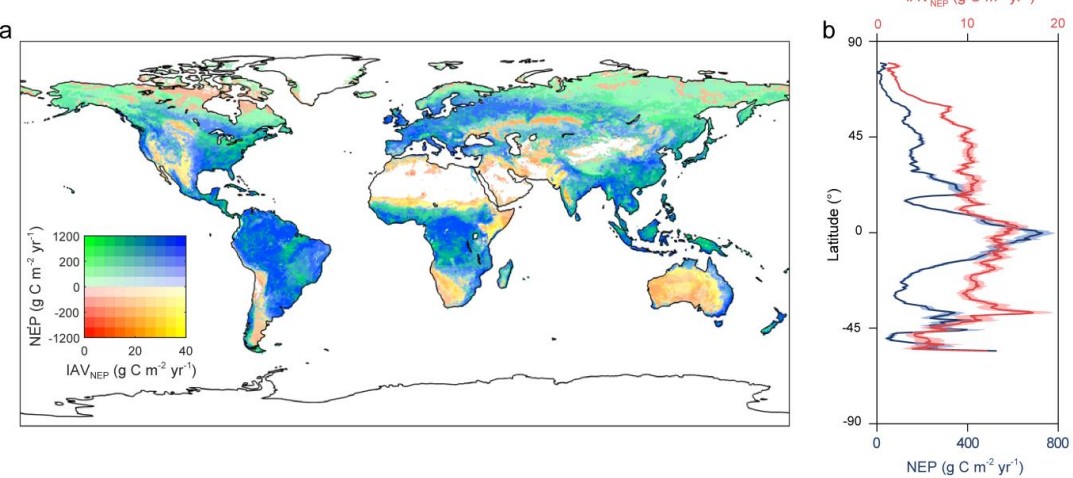


**Figure 1** Locations of carbon sinks (mean annual NEP) and their stability ($IAV_{NEP}$) on the land.

**a**, Spatial patterns of mean annual NEP and $IAV_{NEP}$. **b**, Latitudinal patterns of mean annual NEP and $IAV_{NEP}$.






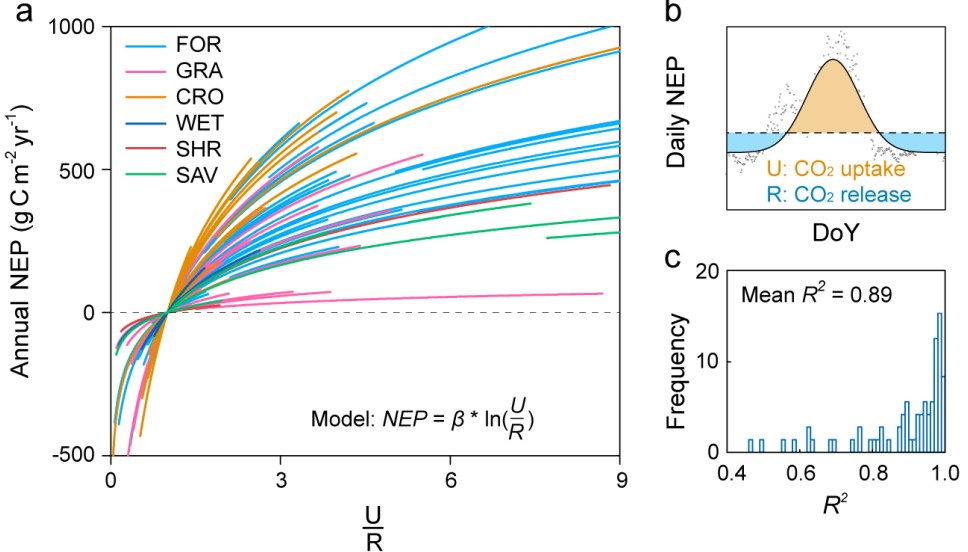


**Figure 2** Relationship between annual NEP and $\frac{U}{R}$ for 72 FLUXNET sites (of the form

NEP $= \beta \cdot \ln\left(\frac{U}{R}\right)$). **a**, Dependence of annual NEP on the ratio between total $CO_2$ exchanges

during net uptake ($U$) and release ($R$) periods (i.e., $\frac{U}{R}$). Each line represents one flux site with

at least 5 years of data. **b**, Conceptual figure for the decomposition framework introduced in

this study. Annual NEP can be quantitatively decomposed into the following indicators:

$NEP = U - R$. **c**, Distribution of the explanation of $\frac{U}{R}$ on temporal variability of FLUXNET

NEP ($R^2$) for FLUXNET sites.





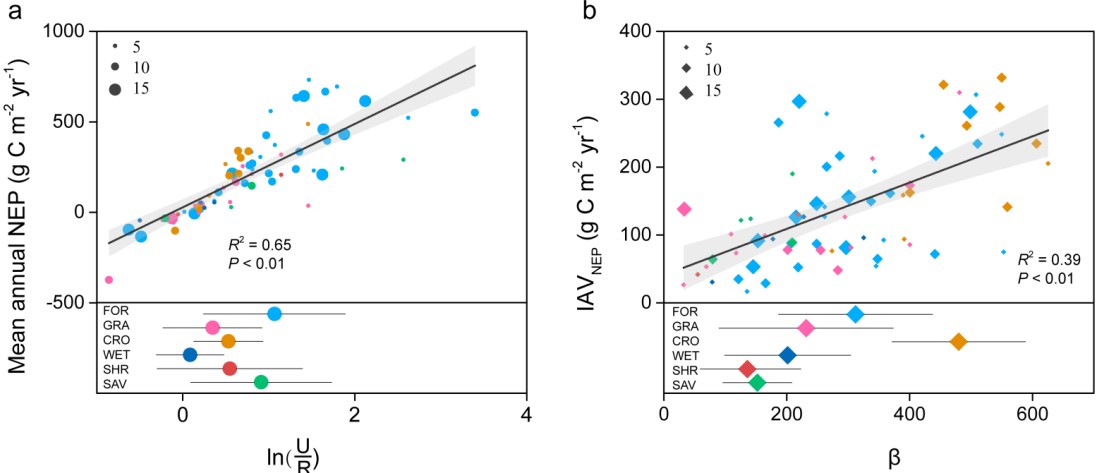

**Figure 3** Contributions of the two indicators in explaining the spatial patterns of mean annual

NEP and IAV$_{NEP}$. **a**, The relationship between annual mean NEP and $\ln\left(\frac{U}{R}\right)$ across

FLUXNET sites ($R^2$ = 0.65, $P$ < 0.01). The insets show the variation of $\ln\left(\frac{U}{R}\right)$ for different

terrestrial biomes. **b**, The explanation of $\beta$ on IAV$_{NEP}$ ($R^2$ = 0.39, $P$ < 0.01). The insets show

the distribution of parameter $\beta$ for different terrestrial biomes. The number of site-years at

each site is indicated with the size of the point.



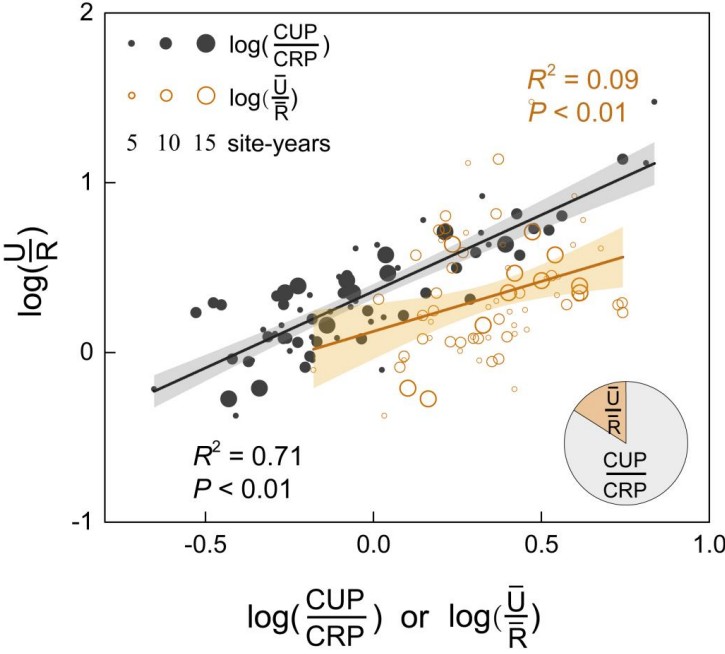

**Figure 4** The linear regression between $\frac{U}{R}$ with $\frac{CUP}{CRP}$ ($R^2 = 0.71$, $P < 0.01$) and $\frac{\bar{U}}{\bar{R}}$ ($R^2 = 0.09$, $P < 0.01$) across sites. The insets show the relative contributions of each indicator to the spatial variation of $\frac{U}{R}$. The number of site-years at each site is indicated with the size of the point.





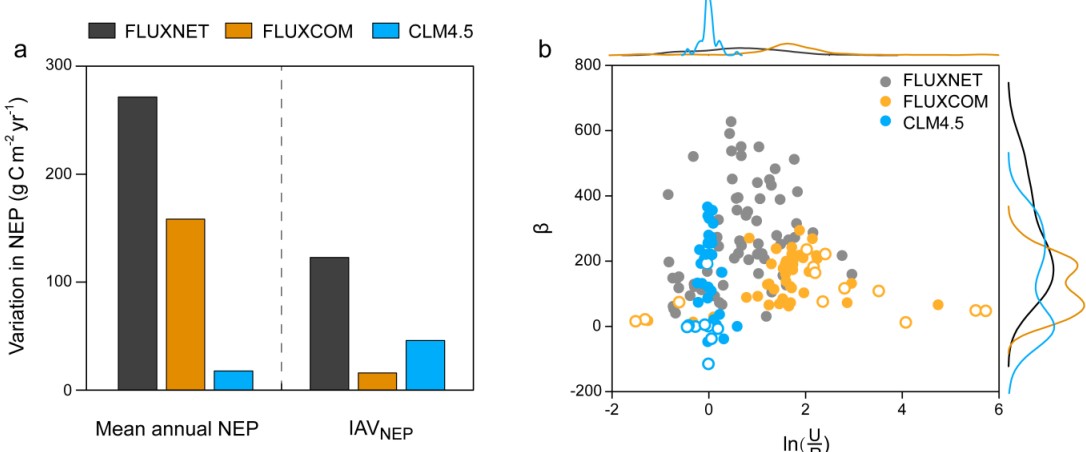

**Figure 5** Representations of the spatially varying NEP and its local indicators in FLUXCOM

product and the Community Land Model (CLM4.5). **a**, The variation of mean annual NEP and

$IAV_{NEP}$ derives from FLUXNET, FLUXCOM and CLM4.5. Variation in mean annual NEP:

the standard deviation of mean annual NEP across sites; Variation in $IAV_{NEP}$: the standard

deviation of $IAV_{NEP}$ across sites. **b**, Representations of the local indicators for NEP in

FLUXNET, FLUXCOM and CLM4.5. The corresponding distributions of $\ln\left(\frac{U}{R}\right)$ and $\beta$ are

shown at the top and right. Significance of the relationship between annual NEP and

$\ln\left(\frac{U}{R}\right)$ for each site is indicated by the circle: closed circles: $P < 0.05$; open circles: $P > 0.05$.

Note that the modeled results are from the pixels extracted from the same locations of the flux

tower sites.



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
