# Peer review of "Research article"

_Biogeosciences, 2020_

## Referee Comment (RC1) · Xiangzhong Luo (Referee) · 23 Mar 2020

General comments: In the manuscript "Spatially asynchronous changes in strength and stability of terrestrial net ecosystem productivity", Chen et al. studied the spatial variations of annual mean NEP and IAV_NEP using in-situ eddy covariance observations and gridded NEP datasets from FLUXCOM and CLM4.5. They proposed a new approach that decomposes NEP into beta, log(U/R) and log (CUP/CRP) and used some of them as "local indicators" to indicate the spatial variation of NEP and IAV_NEP. I am intrigued by this study and find it has the potential to provide some emergent constraints on NEP that we much need at local scales, though I feel some minor revisions are needed to clarify the motivation and the interpretations of the Results.

[Figure]

Specific comments: 1. "Spatially asynchronous" is a bit misleading phrase as it makes me wondering what is meant to be spatially asynchronous/synchronous for NEP, or is it simply used as a substitute for "spatial variation". I think the running title of the manuscript is more accurate which suggests that the authors studied "spatial variability" of NEP and NEP_IAV and found local indicators for them.

2. The first part of the results (section 3.1) serves to prove that there are large spatial variations in NEP and IAV_NEP, and to further motivate a need to study "local indicators" for NEP and IAV_NEP. However, many literatures have reported large spatial variations of NEP and IAV_NEP already, and I feel this kind of reasoning is more suitable to be included in Introduction rather than Results. In addition, FLUXCOM NEP is used here but we know is might not be the best source to study IAV_NEP (Jung et al., 2020).

3. The IAV_NEP and beta for shrublands and savannas are among the smallest compared to other PFTs (Figure 3). Is it at odds with previous global studies that suggest semi-arid ecosystems contributed the most to global IAV_NEP?(Ahlström et al., 2015).

Technical comments: 1. In the legend of Figure 1 please indicate the source of NEP data.

2. L74. Do you mean the "relative differences" between photosynthesis and respiration or between their covariances?

3. L100. Rephrase. "to address the local indicators"?

4. L102. Reference for FLUXNET2015 is Pastorello et al., 2017.

5. L84 -86. Generally, I feel there is a need to clarify why there is a need to find a local indicator (which is also a new phrase)? Does it help in the attribution of spatial variation of NEP and IAV_NEP to different processes, or does it provide an independent constrain on NEP and IAV_NEP?

6. L135. I understand the scale-mismatch between model and eddy-covariance sites is

difficult to address, but is it possible that muted spatial variation of NEP and IAV_NEP from gridded products is partly related to the scale mismatch?

7. L229. "difference" -> "variation".
* * *

---

## Referee Comment (RC2) · Anonymous Referee #2 · 23 Mar 2020

**1. General comments**

Erqian Cui et al. studied the annual NEP and the inter-annual variability of NEP and intended to provide local indicators to better understand their spatial patterns at the FLUXNET site level. I find this study relevant as it is important to have a better understanding of the factors controlling the spatial and inter-annual variability of NEP. However, I have some concerns about some aspects of the method and how the results are presented (see More specific comments section). In addition, there are some results presented in this study that do not provide ay significant new information compared to the available literature (e.g. spatial patterns of annual NEP and IAV of NEP at the global scale). Plus, most of the analysis is done at FLUXNET site level, therefore I do not really the point of using the FLUXCOM and CLM4.5 for the presented study. In

short, although I find the presented study suitable for the scope of Biogeosciences, the manuscript is still in its early stage to be accepted as it is, therefore I suggest to make major revisions before potential acceptance.

2. Specific comments

L. 3-4 The title is very confusing and does not really reflect the findings of the analysis. Please try to rephrase the title so that it matches the message the analysis is trying to convey.

L. 38 "machine-learning-derived database." This concept seems odd and confusing. What about something like "based on a compiled global dataset and a machine learning method". The use "'machine-learning-derived database' is also not entirely true because, as far as I understood, only the FLUXCOM dataset is based on machine-learning approaches. FLUXNET in-situ data and the CLM4.5 product are not using any machine-learning methods.

L. 65 "is related to the strength of carbon sink". It can also relate to the strength of the carbon source. Consider rephrasing to be more generic.

L. 68 Not convinced by the use of 'asynchronously' all over the manuscript, particularly because the results presented in the manuscript do not provide evidence that the spatial patterns of annual NEP or IAV_NEP are not simultaneous or concurrent in time.

L. 76-77 'environmental fluctuations among years'. Musavi et al., 2017 attributed the year-to-year variation to species richness and stand age. In the same line, Besnard et al. 2018 attributed most of the annual NEP variation to forest age.

L. 82-84 Can this sentence be merged with the 1st sentence of the paragraph (L. 71-72)? They seem quite redundant.

L. 84-86 The last sentence of this paragraph seems a bit out of the context of the whole paragraph. Consider improving the transition between the last sentence of the paragraph and the entire paragraph.

L. 85 "could be integrated into some simple indicators". I would use the term 'decompose' instead of 'integrated'. After all, the authors want to decompose the contribution of a series of carbon uptake and carbon release metrics to annual NEP and IAV_NEP.

L. 98-99 Not sure that FLUXCOM products are the best to assess IAV_NEP. Please check Jung et al. 2020 to understand the issues of such products when looking at IAV_NEP. Why not using NEE derived from atmospheric inversions though (e.g. Jena CarboScope (Rödenbeck et al., 2018), CAMSv17r1 (Chevallier et al., 2005, 2019) and CarbonTracker-EU (Peters et al., 2010)). At least, we know that this data capture some processes that contribute to IAV_NEP, which are not being captured with eddy-covariance data (e.g. fire, CO2 fertilization).

L. 122-129 It might be relevant to specify that you use the FLUXCOM RS-meteo products for which the inter-annual variability is only driven by climatic conditions as they used the mean seasonal cycle of remote sensing products. This basically means that there is no inter-annual variability directly related to the state of vegetation.

L. 124 why only using the CRUNCEPv6 product. In my understanding, FLUXCOM uses more than one meteorological forcing as well as different machine-learning methods. Using all the FLUXCOM RS-meteo products could additionally provide uncertainty estimates for the presented indicators.

L. 122-136 If one of the aims is to compare FLUXCOM and CLM4.5, I would suggest comparing the two products during the same time period (i.e. 1990-2010).

L. 133 'match the available FLUXCOM dataset.' Spatially or temporally? As far as I know, the FLUXCOM products have a spatial resolution of either 0.5 or 0.0833 degrees (http://www.fluxcom.org/CF-Products/).

L. 140 equation 1: So U is conceptually GPP and R ecosystem respiration, right? I would be curious to see how GPP compared to U when U is computed as in equation 4 for a sanity check. Are they the same? In principle yes, right? Same for ER and R.

L. 143 I am not sure if this equation is written correctly. Assuming that U is supposed to be expressed in gC m-2 d-1, the way the equation is written suggests that the U would be expressed in gC m-2 (assuming that CUP is a length expressed in the number of days), which is then inconsistent with equation 4. Or did I misunderstand how CUP is calculated?

L. 144 The same applies to this equation.

L. 148-149 I think these equations are correct and good enough to explain how U and R are calculated, therefore I would discard equation (2) and (3) to avoid confusion. Again, U and R derived from equations 2 and 3 do not seem to match how U and R are calculated from eq 4 and 5.

L. 150-153 "Because many studies have [..] are tightly correlated" I would move this sentence to the introduction. I am also not sure that this is enough to justify the need to look at the relationship between annual NEP and the ratio U/R.

L. 160 This equation is correct if one assumes that equations 2 and 3 correct, and if I understood correctly their formulation, equations 2 and 3 are not (see comment above). Therefore, I do not believe that the ratio U/R can be partitioned as presented in equation 7. It seems that part of the paper is based on assuming that equations 2 and 3 are correct, therefore I have concerned related to the analysis relying on equations 2 and 3.

L. 171 I think the analysis presented in section 4 is not correct for the issues I have raised related to equations 2 and 3 at least the way equation 8 is expressed. One could express U/R = f(U/R, CUP/CUR) though and run the variable importance analysis. Why not just do the variable importance analysis as NEP = f(U/R, CUP/CUR)? I find it cleaner although it might be a bit circular and spurious as U and R are derived from NEP.

L. 186 I do not find this section relevant in the context of the study. Besides, most of the

presented results are already well documented in the literature (e.g. Jung at al. 2020).

L. 188 Be aware that the 'large carbon sinks' are very likely related to an artifact in the eddy-covariance datasets due to advection and storage issues. It might be relevant to discuss eddy-covariance data quality issues.

L. 204 Would that make sense to discard the sites for which the logarithmic function does not provide a correlation >0.9 for robustness?

L. 207-208 "This finding suggests that the mean annual ratio ln(U/R) is a good indicator for NEP and its spatial variation." Isn't it expected? I mean U and R are derived from NEP so you might expect that their ratio explains the annual variability of NEP, right?

L. 218 Again, is this analysis being done on the extracted time series for each Fluxnet sites or globally? If the former, I do not really see the point of included results based on FLUXCOM or CLM4.5 for the purpose of the study. It would be interesting to run this analysis both at the global scale and at the Fluxnet level.

L. 219 I do not think that one can directly compare the results from FLUXNET data and the two global products (i.e. FLUXCOM and CLM4.5) simply because of the strong bias in representativeness in the FLUXNET datasets. For instance, there are very few semi-arid ecosystems (e.g. 2 shrublands and 5 savannas in the presented study) in the FLUXNET dataset, while they represent a large portion of land at the global scale and have been shown to substantially control the interannual variability of NEP (Ahlström et al., 2015). Or do you extract FLUXCOM and CLM4.5 time series for each FLUXNET site location? If so, it is anyway not a fair comparison due to spatial mismatch as the footprint of a tower is definitely lower than 1 degree (CLM4.5) or 0.5 degree (FLUXCOM) spatial resolution. As previously mentioned, I would rather run this analysis globally and not only at FLUXNET sites to have a real added value by using global products such as FLUXCOM and CLM4.5.

3. Technical corrections

L.57 'However' does not sound appropriate. Maybe 'furthermore' or 'in addition'.

L. 62 'dramatic'. Try to avoid emotional semantic in a scientific paper. Maybe 'substantial' instead?

L. 77. replace Musavi, 2017 by Musavi et al., 2017

L. 104 'database' Replace database by product.

L. 119-121 Stand age information is mentioned here but is they even being used further in the analysis? If not, please remove it.

L. 154-155 'Then we found that annual NEP [...] (Figure S2).' To me, this already belongs to the results section.

L. 154 'the ratio U/R'. It might be relevant for the reader to see a sentence explaining the meaning of the ratio U/R. This explanation in L. 162-163 comes a bit too late.

L. 151-152 'the non-growing soil respiration' Is that what you mean here? Maybe rephrase.

L. 208 I would not say 'was well explained' but rather that the correlation was moderate (i.e. $0.3 > r > 0.7$)

L. 347 In Fig. 1, it is not clear to me what products are we looking at. FLUXCOM, CLM 4.5 or both? It seems to be FLUXCOM (L. 99) but please specify in the figure's caption.

References:

Ahlström, Anders, et al. "The dominant role of semi-arid ecosystems in the trend and variability of the land $CO_2$ sink." Science 348.6237 (2015): 895-899.

Besnard, Simon, et al. "Quantifying the effect of forest age in annual net forest carbon balance."Environmental Research Letters 13.12 (2018): 124018.

Chevallier, F., et al. "Inferring $CO_2$ sources and sinks from satellite observations: Method and application to TOVS data."Journal of Geophysical Research: Atmospheres110.D24 (2005).

Chevallier, Frédéric, et al. "Objective evaluation of surface-and satellite-driven carbon dioxide atmospheric inversions."Atmospheric Chemistry and Physics 19.22 (2019): 14233-14251.

Jung, Martin, et al. "Scaling carbon fluxes from eddy covariance sites to globe: Synthesis and evaluation of the FLUXCOM approach."Biogeosciences17.5 (2020): 1343-1365.

Peters, W., et al. "Seven years of recent European net terrestrial carbon dioxide exchange constrained by atmospheric observations."Global Change Biology 16.4 (2010): 1317-1337.

Rödenbeck, Christian, et al. "How does the terrestrial carbon exchange respond to inter-annual climatic variations?: A quantification based on atmospheric CO2 data."Biogeosciences (2018).

---

## Author Comment (AC1) · 9 May 2020

Response to comments from reviewer #2 Dear editor: Thank you very much for handling our manuscript. We really appreciate the reviewer for the invaluable suggestions and comments on our manuscript. Below, we address all the comments from reviewer #2 point-by-point. The comments are italicized and our response follow in blue, and we hope we could address the concerns from reviewer. Reply to Reviewer #2 General comments: Comment 1B: Erqian Cui et al. studied the annual NEP and the inter-annual variability of NEP and intended to provide local indicators to better understand their spatial patterns at the FLUXNET site level. I find this study relevant as it is important to have a better understanding of the factors controlling the spatial and inter-annual variability of NEP. However, I have some concerns about some aspects of the method

and how the results are presented (see More specific comments section). In addition, there are some results presented in this study that do not provide ay significant new information compared to the available literature (e.g. spatial patterns of annual NEP and IAV of NEP at the global scale). Plus, most of the analysis is done at FLUXNET site level, therefore I do not really the point of using the FLUXCOM and CLM4.5 for the presented study. In short, although I find the presented study suitable for the scope of Biogeosciences, the manuscript is still in its early stage to be accepted as it is, therefore I suggest to make major revisions before potential acceptance. Response: Thanks for the valuable suggestions. Based the reviewer's comment, we have made a substantial revision on both of the Method and Results sections. First, we have deleted Figure 1 from Results, and moved the related content to the Introduction Section as the background of our study. Second, we have showed the major findings with FLUXNET observations and the atmospheric inversion product (i.e. new results in Figure 1B). Then as suggested by the reviewer, we have benchmarked the simulations from the compiled global product and the process-based global model both at the global scale and at the FLUXNET site level (i.e. new results in Figure 4B). Specific comments: Comment 2B: L. 3-4 The title is very confusing and does not really reflect the findings of the analysis. Please try to rephrase the title so that it matches the message the analysis is trying to convey. Response: Thanks, we have revised the title as "Spatial variations in terrestrial net ecosystem productivity and its local indicators" Comment 3B: L. 38 "machine-learning-derived database." This concept seems odd and confusing. What about something like "based on a compiled global dataset and a machine learning method". The use "'machine-learning-derived database' is also not entirely true because, as far as I understood, only the FLUXCOM dataset is based on machine learning approaches. FLUXNET in-situ data and the CLM4.5 product are not using any machine-learning methods. Response: We have rephrased the relevant statement as "the compiled global product and the process-based global model.". Comment 4B: L. 65 "is related to the strength of carbon sink". It can also relate to the strength of the carbon source. Consider rephrasing to be more generic. Response: Thanks, we

have rephrased this sentence as "is related to the strength of carbon exchange" . Comment 5B: L. 68 Not convinced by the use of 'asynchronously' all over the manuscript, particularly because the results presented in the manuscript do not provide evidence that the spatial patterns of annual NEP or IAV_NEP are not simultaneous or concurrent in time. Response: Thanks, we have deleted the word "asynchronously" all over the manuscript and replaced it with "variation". Comment 6B: L. 76-77 'environmental fluctuations among years'. Musavi et al., 2017 attributed the year-to-year variation to species richness and stand age. In the same line, Besnard et al. 2018 attributed most of the annual NEP variation to forest age. Response: Thanks, we have revised this sentence as "Many previous analyses have attributed the IAVNEP at the site level to the different sensitivities of ecosystem photosynthesis and respiration to environmental drivers (Gilmanov et al., 2005; Reichstein et al., 2005) and biotic controls (Besnard et al., 2018; Musavi et al., 2017).". Comment 7B: L. 82-84 Can this sentence be merged with the 1st sentence of the paragraph (L.71-72)? They seem quite redundant. Response: Thanks. Sorry for the misunderstanding of these two sentences. The first sentence illustrates the decomposition of NEP as the difference between photosynthesis and respiration, while the last sentence leads to the decomposition of NEP directly into $CO_2$ uptake flux and $CO_2$ release flux. To make these points clearer, we have rephrased this sentence as: "Alternatively, the annual NEP of a given ecosystem can be also directly decomposed into $CO_2$ uptake flux and $CO_2$ release flux (Gray et al., 2014), which are more direct components for NEP (Fu et al., 2019). It is still unclear whether ecosystem $CO_2$ uptake and release fluxes could be attributed to some simple indicators for the spatially varying NEP and IAVNEP in terrestrial ecosystems." Comment 8B: L. 84-86 The last sentence of this paragraph seems a bit out of the context of the whole paragraph. Consider improving the transition between the last sentence of the paragraph and the entire paragraph. Response: Thanks, we have rephrased this section and strengthened our points by adding the following sentences: "However, despite the previous efforts in a predictive understanding of the land-atmospheric C exchanges, the multi-model spread has not changed over time (Arora et al., 2019).

Therefore, it is imperative to explore the potential indicators for the spatially varying NEP, which could help attribute the spatial variation of NEP and IAVNEP into different processes and provide valuable constraints for the global C cycle. Alternatively, the annual NEP of a given ecosystem can be also directly decomposed into CO2 uptake flux and CO2 release flux (Gray et al., 2014), which are more direct components for NEP (Fu et al., 2019). It is still unclear whether ecosystem CO2 uptake and release fluxes could be attributed to the spatially varying NEP and IAVNEP in terrestrial ecosystems." Comment 9B: L. 85 "could be integrated into some simple indicators". I would use the term 'decompose' instead of 'integrated'. After all, the authors want to decompose the contribution of a series of carbon uptake and carbon release metrics to annual NEP and IAV_NEP. Response: Thanks, done as suggested. Comment 10B: L. 98-99 Not sure that FLUXCOM products are the best to assess IAV_NEP. Please check Jung et al. 2020 to understand the issues of such products when looking at IAV_NEP. Why not using NEE derived from atmospheric inversions though (e.g. JenaCarboScope (Rödenbeck et al., 2018), CAMSv17r1 (Chevallier et al., 2005, 2019) and CarbonTracker-EU (Peters et al., 2010)). At least, we know that this data capture some processes that contribute to IAV_NEP, which are not being captured with eddy-covariance data (e.g. fire, CO2 fertilization). Response: Thanks for the suggestion. We have verified the relationship derived from FLUXNET sites with the Jena CarboScope CO2 Inversion, and find that the relationship between annual NEP and U/R is robust in most global grid cells. We have added these new analyses in the Results Section and Figure 2 to strengthen our findings: "In addition, the relationship between NEP and U/R was also verified by the atmospheric inversion product (i.e., Jena CarboScope Inversion). The control of U/R on annual NEP was robust in most global grid cells (i.e. $0.6 < R2 < 1$). The explanation of U/R was higher in 80% of the regions, but lower in North American (Fig. 2). These two datasets both showed that the indicator U/R could successfully capture the variability in annual NEP."

Figure 1B. Relationship between annual NEP and U/R for Jena Inversion product (of the form NEP=$\beta$·lnâĄą(U/R)). The black box indicates the location of the sample. Comment 11B: L. 122-129 It might be relevant to specify that you use the FLUXCOM RS-meteo products for which the inter-annual variability is only driven by climatic conditions as they used the mean seasonal cycle of remote sensing products. This basically means that there is no inter-annual variability directly related to the state of vegetation. Response: Thanks, we have rephrased the description of FLUXCOM product by adding the following sentences in Method Section (Lines x-x): "It should be noted that the inter-annual variability of FLUXCOM product is only driven by climatic conditions, the effects of land use and land cover change are not represented." Comment 12B: L. 124 why only using the CRUNCEPv6 product. In my understanding, FLUXCOM uses more than one meteorological forcing as well as different machine-learning methods. Using all the FLUXCOM RS-meteo products could additionally provide uncertainty estimates for the presented indicators. Response: Thanks for this comment. We used the CRUNCEPv6 product mainly due to two reasons. First, the simulations from CLM4.5 and Jena Inversion in this study are both driven by CRUNECP meteorological forcing. Therefore, in order to reduce the uncertainty caused by meteorological forcing, we would prefer to choose the CRUNCEPv6 product. Second, we have averaged all the FLUXCOM CRUNCEPv6 products with different machine-learning methods to avoid the uncertainty caused by machine-learning methods. To illustrate our consideration clearer, we have detailed the selection of the product in Method Section (Lines x-x): "To be consistent with the meteorological forcing of Jena Inversion product and the CLM4.5 model, we used the FLUXCOM CRUNCEPv6 products. In addition, in order to reduce the uncertainty caused by machine-learning methods, we averaged all the FLUXCOM CRUNCEPv6 products with different machine-learning methods." Comment 13B: L. 122-136 If one of the aims is to compare FLUXCOM and CLM4.5, I would suggest comparing the two products during the same time period (i.e. 1990-2010). Response: Thanks for this suggestion, we have adjusted the time period of all the global products to 1985-2010. Comment 14B: L. 133 'match the available FLUXCOM dataset.' Spatially or temporally? As far as I know, the FLUXCOM products have a spatial resolution of either 0.5 or 0.0833 degrees (http://www.fluxcom.org/CF-Products/).

Response: Thanks, we have adjusted the global products to the same time period and specified their spatial resolution in the Method Section. Comment 15B: L. 140 equation 1: So U is conceptually GPP and R ecosystem respiration, right? I would be curious to see how GPP compared to U when U is computed as in equation 4 for a sanity check. Are they the same? In principle yes, right? Same for ER and R. Response: Sorry for the misunderstanding. We have drawn a concept figure to shown the decomposition of NEP in our study. The annual NEP is determined by vegetation photosynthesis and ecosystem respiration, but here we decompose the annual NEP into its more direct components: CO2 uptake flux and CO2 release flux. To describe the decomposition process more clearly, we have modified the decomposition process of NEP in Method Section.

Figure 2B. Conceptual figure for the decomposition of annual NEP in this study. The example shows daily observations from BE-Bra site. Comment 16B: L. 143 I am not sure if this equation is written correctly. Assuming that U is supposed to be expressed in gC m-2 d-1, the way the equation is written suggests that the U would be expressed in gC m-2 (assuming that CUP is a length expressed in the number of days), which is then inconsistent with equation 4. Or did I misunderstand how CUP is calculated? Response: Thanks for reminding the confusion of the units. In this study, U is expressed in gC m-2 and calculated from the mean daily CO2 uptake (U ÌЕ, gC m-2 d-1) over the carbon uptake period (CUP, d). Actually, the results of equations (2)-(3) and equations (4)-(5) are mathematically equivalent. However, as suggested by the reviewer, the units of these two approaches are ambiguous. Therefore, considering the subsequent analysis, we have deleted equations (4) and (5). Comment 17B: L. 144 The same applies to this equation. Response: Thanks, we have deleted equations (4) and (5). Comment 18B: L. 148-149 I think these equations are correct and good enough to explain how U and R are calculated, therefore I would discard equation (2) and (3) to avoid confusion. Again, U and R derived from equations 2 and 3 do not seem to match how U and R are calculated from eq 4 and 5. Response: Thanks. We have discard equations (2) and (3) to avoid

confusion Comment 19B: L. 150-153 "Because many studies have [..] are tightly correlated" I would move this sentence to the introduction. I am also not sure that this is enough to justify the need to look at the relationship between annual NEP and the ratio U/R. Response: Thanks for this suggestion, we have removed these sentences to the Introduction Section and added several sentences to state the motivation to explore the relationship between annual NEP and its components U and R: "However, despite the previous efforts in a predictive understanding of the land-atmospheric C exchanges, the multi-model spread has not changed over time (Arora et al., 2019). Therefore, it is imperative to explore the potential indicators for the spatially varying NEP, which could help attribute the spatial variation of NEP and IAVNEP into different processes and provide valuable constraints for the global C cycle. Alternatively, the annual NEP of a given ecosystem can be also directly decomposed into $CO_2$ uptake flux and $CO_2$ release flux (Gray et al., 2014), which are more direct components for NEP (Fu et al., 2019). Many studies have reported that the vegetation $CO_2$ uptake during the growing season and the non-growing season soil respiration are tightly correlated (Luo et al., 2014; Zhao et al., 2016). It is still unclear how the ecosystem $CO_2$ uptake and release fluxes would control the spatially varying NEP." Comment 20B: L. 160 This equation is correct if one assumes that equations 2 and 3 correct, and if I understood correctly their formulation, equations 2 and 3 are not (see comment above). Therefore, I do not believe that the ratio U/R can be partitioned as presented in equation 7. It seems that part of the paper is based on assuming that equations 2 and 3 are correct, therefore I have concerned related to the analysis relying on equations 2 and 3. Response: Thanks for this comment. To be consistent with the equation (7), we have deleted the equations (4) and (5) and kept the equations (2) and (3) as the final decomposition approaches. Comment 21B: L. 171 I think the analysis presented in section 4 is not correct for the issues I have raised related to equations 2 and 3 at least the way equation 8 is expressed. One could express U/R = f(U/R, CUP/CUR) though and run the variable importance analysis. Why not just do the variable importance analysis as NEP = f(U/R, CUP/CUR)? I find it cleaner although

it might be a bit circular and spurious as U and R are derived from NEP. Response: Thanks for this valuable suggestion. In the revised version, we have directly tested the effect of these two ratios on the spatial variation in NEP (Figure 3B). These new results have been added in the Results as Figure 4. The major revisions in Method Section and Results Section are as below: Method Section: "We further quantified the relative contributions of U ÌĚ/R ÌĚ and CUP/CRP in driving the spatial variations in NEP: NEP=$\int (U/R, CUP/CRP) (6) We used a relative importance analysis method to quantify the relative contributions of each ratio$

*"The decomposition of indicator U/R into U/R and CUP/CRP allowed us to quantify the relative importance of these two ratios in site variation of NEP (Fig. 4). Therefore, the spatial distribution of mean annual NEP was mostly driven by the phenological ra$

Figure 3B. The relative contributions of the local indicators in explaining the spatial patterns of mean annual NEP. a, The linear regression between mean annual NEP with CUP/CRP ($R^2$ = 0.33, $P < 0.01$) and U ÌĚ/R ÌĚ ($R^2$ = 0.25, $P < 0.01$) across sites. b, The relative contributions of each indicator to the spatial variation of NEP. The number of site-years at each site is indicated with the size of the point. Comment 22B: L. 186 I do not find this section relevant in the context of the study. Besides, most of the presented results are already well documented in the literature (e.g. Jung at al. 2020). Response: Thanks for this suggestion. We have deleted this section from Results, and moved the related content to the Introduction Section: "Large spatial difference in terrestrial NEP has been reported from eddy-flux measurements, model outputs and atmospheric inversion products. In addition, the global average IAV of NEP was large relative to global annual mean NEP (Baldocchi et al., 2018). More importantly, the spatial variations of NEP and IAVNEP were typically underestimated by the compiled global dataset and the process-based global models (Jung et al., 2020; Fu et al., 2019)." Comment 23B: L. 188 Be aware that the 'large carbon sinks' are very likely related to an artifact in the eddy-covariance datasets due to advection and storage issues. It might be relevant to discuss eddy-covariance data quality issues. Response: Thanks for this suggestion, and this section has been removed. Comment 24B: L. 204 Would that make sense to discard the sites for which the logarithmic function does not provide a correlation >0.9 for robustness? Response: Thanks, we have rephrased this sentence as "The logarithmic correlations between annual NEP and U/R were significant at all sites (Fig. 1a; Fig. S2), and ∼90% of R2 falling within a range from 0.7 to 1 (Fig. 1c)." Comment 25B: L. 207-208 "This finding suggests that the mean annual ratio ln(U/R) is a good indicator for NEP and its spatial variation." Isn't it expected? I mean U and R are derived from NEP so you might expect that their ratio explains the annual variability of NEP, right? Response: Thanks. We have rephrased the related sentences to make the statements clearer: (1) Results Section 3.1: "These two datasets both showed that the indicator U/R could successfully capture the variability in annual NEP." (2) Results Section 3.2: "This finding suggested that the mean annual ratio lnâĄą(U/R) is a good indicator for cross-site variation in NEP." Comment 26B: L. 218 Again, is this analysis being done on the extracted time series for each Fluxnet sites or globally? If the former, I do not really see the point of included results based on FLUXCOM or CLM4.5 for the purpose of the study. It would be interesting to run this analysis both at the global scale and at the Fluxnet level. Response: Thanks for this valuable suggestion. We have done additional analyses at the global scale: First, yes, the previous analysis in Figure 5 is based on the extracted time series for FLUXNET sites. Second, as suggested by the reviewer, we also have run the same analysis at the global scale based on Jena Inversion product, FLUXCOM product and CLM4.5 model (Figure 4B). The results have strengthened our major conclusion that the spatial variation of mean annual NEP can be indicated by ln(U/R), while the spatial distribution of IAVNEP is well indicated by the slope (i.e., $\beta$) of the demonstrated logarithmic correlation. We have added these new analyses in Results Section as Figure 6. The major revisions in Results Section are as below: "However, the spatial variations of NEP and IAVNEP were associated with the spatial resolution of the product (Marcolla et al., 2017). At the global scale, the spatial variation of mean annual NEP can be also well indicated by ln(U/R) (Fig. 6). The widely reported larger C uptake in FLUXCOM (Jung et al., 2020) resulted from its higher simulations for U/R. In addition, the larger spatial variation of IAVNEP in CLM4.5 could be inferred from the indicator $\beta$."

Figure 4B. Representations of the spatially varying NEP and its local indicators in

FLUXCOM product and the Community Land Model (CLM4.5) at the global scale. a, The variation of mean annual NEP and IAVNEP derives from Jena Inversion, FLUXCOM and CLM4.5. Variation in mean annual NEP: the spatial variation of mean annual NEPs; Variation in IAVNEP: the spatial variation of standard deviation in IAVNEP. b, Representations of the local indicators for NEP in Jena Inversion, FLUXCOM and CLM4.5. Comment 27B: L. 219 I do not think that one can directly compare the results from FLUXNET data and the two global products (i.e. FLUXCOM and CLM4.5) simply because of the strong bias in representativeness in the FLUXNET datasets. For instance, there are very few semi-arid ecosystems (e.g. 2 shrublands and 5 savannas in the presented study) in the FLUXNET dataset, while they represent a large portion of land at the global scale and have been shown to substantially control the interannual variability of NEP (Ahlström et al., 2015). Or do you extract FLUXCOM and CLM4.5 time series for each FLUXNET site location? If so, it is anyway not a fair comparison due to spatial mismatch as the footprint of a tower is definitely lower than 1 degree (CLM4.5) or 0.5 degree (FLUXCOM) spatial resolution. As previously mentioned, I would rather run this analysis globally and not only at FLUXNET sites to have a real added value by using global products such as FLUXCOM and CLM4.5. Response: Thanks for the comment on scale mismatch. As suggested by the reviewer, we have done the same analysis both at the global scale and at the FLUXNET site level. The results from FLUXNET sites are used to benchmark the simulations of FLUXCOM product and CLM4.5 model at the FLUXNET site level, and the results from Jena Inversion product are used to evaluate the simulations of FLUXCOM product and CLM4.5 model at the global scale. As shown in Figure 4B, the analyses at the global scale and at the FLUXNET site level both support our major conclusion that the spatial variation of mean annual NEP can be indicated by ln(U/R), while the spatial distribution of IAVNEP is well indicated by the slope (i.e., $\beta$) of the demonstrated logarithmic correlation. Technical corrections: Comment 28B: L.57 'However' does not sound appropriate. Maybe 'furthermore' or 'in addition'. L. 62 'dramatic'. Try to avoid emotional semantic in a scientific paper. Maybe 'substantial' instead? L. 77.

replace Musavi, 2017 by Musavi et al., 2017 L. 104 'database' Replace database by product. Response: Done as suggested. L. 119-121 Stand age information is mentioned here but is they even being used further in the analysis? If not, please remove it. Response: Removed. L. 154-155 'Then we found that annual NEP [...] (Figure S2).' To me, this already belongs to the results section. Response: Thanks, we have removed this sentence to the Results Section. L. 154 'the ratio U/R'. It might be relevant for the reader to see a sentence explaining the meaning of the ratio U/R. This explanation in L. 162-163 comes a bit too late. Response: Thanks, we have added the meaning of ratio U/R as "we further tested the relationship between annual NEP and the ratio of U/R. Ecologically, the ratio of U/R reflects the relative strength of the ecosystem $CO_2$ uptake.". L. 151-152 'the non-growing soil respiration' Is that what you mean here? Maybe rephrase. Response: Thanks, we have rephrased it as "the non-growing season soil respiration". L. 208 I would not say 'was well explained' but rather that the correlation was moderate (i.e. $0.3 > r > 0.7$) Response: Thanks, we have rephrased it as "was moderately explained". L. 347 In Fig. 1, it is not clear to me what products are we looking at. FLUXCOM, CLM 4.5 or both? It seems to be FLUXCOM (L. 99) but please specify in the figure's caption. Response: As suggested by Comment 22B, we have deleted Figure 1 and the related results. References: Ahlström, Anders, et al. "The dominant role of semi-arid ecosystems in the trend and variability of the land $CO_2$ sink." Science 348.6237 (2015): 895-899. Besnard, Simon, et al. "Quantifying the effect of forest age in annual net forest carbon balance." Environmental Research Letters 13.12 (2018): 124018. Jung, Martin, et al. "Scaling carbon fluxes from eddy covariance sites to globe: Synthesis and evaluation of the FLUXCOM approach." Biogeosciences17.5 (2020): 1343-1365. Marcolla, B., Rödenbeck, C., and Cescatti, A.: Patterns and controls of inter-annual variability in the terrestrial carbon budget. Biogeosciences, 14, 3815-3829, 2017. Rödenbeck, Christian, et al. "How does the terrestrial carbon exchange respond to inter-annual climatic variations?: A quantification based on atmospheric $CO_2$ data." Biogeosciences (2018).

Please also note the supplement to this comment:
https://www.biogeosciences-discuss.net/bg-2020-26/bg-2020-26-AC1-supplement.pdf

[Figure]

[Figure]

**Fig. 1.** Relationship between annual NEP and U/R for Jena Inversion product (of the form NEP=$\beta$·lnâĄą(U/R)).

[Figure]

Fig. 2. Conceptual figure for the decomposition of annual NEP in this study

a

Mean annual NEP (g C m$^{-2}$ yr$^{-1}$)

1000

500

0

-500

$\ln(\frac{CUP}{CRP})$

$\ln(\frac{\bar{U}}{R})$

5  10  15

$R^2 = 0.25$
$P < 0.01$

$R^2 = 0.33$
$P < 0.01$

-1          0          1          2

$\ln(\frac{CUP}{CRP})$  or  $\ln(\frac{\bar{U}}{R})$

b

Relative importance (%)

60

40

20

0

$\ln(\frac{CUP}{CRP})$  $\ln(\frac{\bar{U}}{R})$

**Fig. 3.** The relative contributions of the local indicators in explaining the spatial patterns of mean annual NEP.

[Figure]

**Fig. 4.** Representations of the spatially varying NEP and its local indicators in FLUXCOM product and the Community Land Model (CLM4.5) at the global scale

---

## Author Comment (AC2) · 9 May 2020

Response to comments from reviewer #1 Dear editor: Thank you very much for handling our manuscript. We really appreciate the reviewer's insightful comments and suggestions. Below, we address the comments from reviewer #1 point-by-point. The comments are italicized and our response follow in blue, and we hope we could address the concerns from reviewer. Reply to Reviewer #1 General comments: Comment 1A: In the manuscript "Spatially asynchronous changes in strength and stability of terrestrial net ecosystem productivity", Chen et al. studied the spatial variations of annual mean NEP and IAV_NEP using in-situ eddy covariance observations and gridded NEP datasets from FLUXCOM and CLM4.5. They proposed a new approach that decomposes NEP into beta, log(U/R) and log (CUP/CRP) and used some of them as "local

indicators" to indicate the spatial variation of NEP and IAV_NEP. I am intrigued by this study and find it has the potential to provide some emergent constraints on NEP that we much need at local scales, though I feel some minor revisions are needed to clarify the motivation and the interpretations of the Results. Response: Thanks for the recognitions and valuable suggestions. Specific comments: Comment 2A: "Spatially asynchronous" is a bit misleading phrase as it makes me wondering what is meant to be spatially asynchronous/synchronous for NEP, or is it simply used as a substitute for "spatial variation". I think the running title of the manuscript is more accurate which suggests that the authors studied "spatial variability" of NEP and NEP_IAV and found local indicators for them. Response: Thanks, we have revised the title as "Spatial variations in terrestrial net ecosystem productivity and its local indicators". Comment 3A: The first part of the results (section 3.1) serves to prove that there are large spatial variations in NEP and IAV_NEP, and to further motivate a need to study "local indicators" for NEP and IAV_NEP. However, many literatures have reported large spatial variations of NEP and IAV_NEP already, and I feel this kind of reasoning is more suitable to be included in Introduction rather than Results. In addition, FLUXCOM NEP is used here but we know is might not be the best source to study IAV_NEP (Jung et al., 2020). Response: Thanks for this suggestion. We have deleted this part of results, and moved the related content to the Introduction Section: "Large spatial difference in terrestrial NEP has been reported from eddy-flux measurements, model outputs and atmospheric inversion products. In addition, the global average IAV of NEP was large relative to global annual mean NEP (Baldocchi et al., 2018). More importantly, the spatial variations of NEP and IAVNEP were typically underestimated by the compiled global dataset and the process-based global models (Jung et al., 2020; Fu et al., 2019)." Comment 4A: The IAV_NEP and beta for shrublands and savannas are among the smallest compared to other PFTs (Figure 3). Is it at odds with previous global studies that suggest semi-arid ecosystems contributed the most to global IAV_NEP? (Ahlström et al., 2015). Response: Thanks for this suggestion. As the reviewer has mentioned, there are very few semi-arid ecosystems (e.g. 2 shrublands and 5 savannas in the presented study)

in the FLUXNET sites, while they represent a large portion of land at the global scale and have been shown to substantially control the interannual variability of NEP. Therefore, we have added several sentences in Discussion Section to illustrate this point: "However, the relatively lower $\beta$ in shrublands and savannas should be interpreted cautiously. There are very few semi-arid ecosystems in the FLUXNET sites, while they represent a large portion of land at the global scale and have been shown to substantially control the interannual variability of NEP (Ahlström et al., 2015)." Technical comments: Comment 5A: In the legend of Figure 1 please indicate the source of NEP data. Response: This section has been removed. Comment 6A: L74. Do you mean the "relative differences" between photosynthesis and respiration or between their covariances? Response: Thanks, we have rephrased this sentence as "Because photosynthesis and respiration are strongly correlated over space (Baldocchi et al., 2015; Biederman et al., 2016), their relative difference could determine the spatial variation of NEP." Comment 7A: L100. Rephrase. "to address the local indicators"? Response: Thanks, we have rephrased this sentence as "In this study, we decomposed annual NEP into U and R, and explored the local indicators for spatially varying NEP." Comment 8A: L102. Reference for FLUXNET2015 is Pastorello et al., 2017. Response: Thanks, revised. Comment 9A: L84 -86. Generally, I feel there is a need to clarify why there is a need to find a local indicator (which is also a new phrase)? Does it help in the attribution of spatial variation of NEP and IAV_NEP to different processes, or does it provide an independent constrain on NEP and IAV_NEP? Response: Thanks for this valuable suggestion. The suggestion proposed by the reviewer inspires us to reorganize the importance of our work. We have added several sentences in Introduction Section to state the necessary of exploring the local indicators: "However, despite the previous efforts in a predictive understanding of the land-atmospheric C exchanges, the multi-model spread has not changed over time (Arora et al., 2019). Therefore, it is imperative to explore the potential indicators for the spatially varying NEP, which could help attribute the spatial variation of NEP and IAVNEP into different processes and provide valuable constraints for the global C cycle." Comment 10A: L135. I understand

the scale-mismatch between model and eddy-covariance sites is difficult to address, but is it possible that muted spatial variation of NEP and IAV_NEP from gridded products is partly related to the scale mismatch? Response: Thanks for this suggestion. Considering the scale mismatch between FLUXNET sites and the gridded product, we have run the same analysis at the global scale based on Jena Inversion product, FLUX-COM product and CLM4.5 model (Figure 1A). The results have strengthened our major conclusion that the spatial variation of mean annual NEP can be indicated by ln(U/R), while the spatial distribution of IAVNEP is well indicated by the slope (i.e., $\beta$) of the demonstrated logarithmic correlation. We have added these new analyses in Results Section as Figure 6. The major revisions in Results Section are as below: "However, the spatial variations of NEP and IAVNEP were associated with the spatial resolution of the product (Marcolla et al., 2017). At the global scale, the spatial variation of mean annual NEP can be also well indicated by ln(U/R) (Fig. 6). The widely reported larger C uptake in FLUXCOM (Jung et al., 2020) resulted from its higher simulations for U/R. In addition, the larger spatial variation of IAVNEP in CLM4.5 could be inferred from the indicator $\beta$."

Figure 1A. Representations of the spatially varying NEP and its local indicators in FLUXCOM product and the Community Land Model (CLM4.5) at the global scale. a, The variation of mean annual NEP and IAVNEP derives from Jena Inversion, FLUXCOM and CLM4.5. Variation in mean annual NEP: the spatial variation of mean annual NEPs; Variation in IAVNEP: the spatial variation of standard deviation in IAVNEP. b, Representations of the local indicators for NEP in Jena Inversion, FLUXCOM and CLM4.5. Comment 11A: L229. "difference" -> "variation". Response: Done as suggested.

Please also note the supplement to this comment:
https://www.biogeosciences-discuss.net/bg-2020-26/bg-2020-26-AC2-supplement.pdf

[Figure]

**Fig. 1.** Representations of the spatially varying NEP and its local indicators in FLUXCOM product and the Community Land Model (CLM4.5) at the global scale.

---

## Author Response (AR1)

<h1 align="center">Response to comments from reviewer #1</h1>

Dear editor:

Thank you very much for handling our manuscript. We really appreciate the reviewer's insightful comments and suggestions. Below, we address the comments from reviewer #1 point-by-point. The comments are italicized and our response follow in blue, and we hope we could address the concerns from reviewer.

**Reply to Reviewer #1**

**General comments:**

**Comment 1A:** *In the manuscript "Spatially asynchronous changes in strength and stability of terrestrial net ecosystem productivity", Chen et al. studied the spatial variations of annual mean NEP and IAV_NEP using in-situ eddy covariance observations and gridded NEP datasets from FLUXCOM and CLM4.5. They proposed a new approach that decomposes NEP into beta, log(U/R) and log (CUP/CRP) and used some of them as "local indicators" to indicate the spatial variation of NEP and IAV_NEP. I am intrigued by this study and find it has the potential to provide some emergent constraints on NEP that we much need at local scales, though I feel some minor revisions are needed to clarify the motivation and the interpretations of the Results.*

Response: Thanks for the recognitions and valuable suggestions. The comments from the reviewer have inspired us to strengthen the importance of the local indicators. We have added one sentence in *Introduction Section* (Lines 83-86) to extend the motivation of this study:

> "Therefore, it is imperative to explore the potential indicators for the spatially varying NEP, which could help attribute the spatial variation of NEP and $IAV_{NEP}$ into different processes and provide valuable constraints for the global C cycle."

**Specific comments:**

**Comment 2A:** *"Spatially asynchronous" is a bit misleading phrase as it makes me wondering what is meant to be spatially asynchronous/synchronous for NEP, or is it simply used as a substitute for "spatial variation". I think the running title of the manuscript is more accurate which suggests that the authors studied "spatial variability" of NEP and NEP_IAV and found local indicators for them.*

Response: Thanks, we have revised the title as "Spatial variations in terrestrial net ecosystem productivity and its local indicators".

**Comment 3A:** *The first part of the results (section 3.1) serves to prove that there are large spatial variations in NEP and IAV_NEP, and to further motivate a need to study "local indicators" for NEP and IAV_NEP. However, many literatures have reported large spatial variations of NEP and IAV_NEP already, and I feel this kind of reasoning is more suitable to be included in Introduction rather than Results. In addition, FLUXCOM NEP is used here but we know is might not be the best source to study IAV_NEP (Jung et al., 2020).*

Response: Thanks for this suggestion. We have deleted this part of results, and moved the related content to the *Introduction Section* (Lines 65-69):

> "Large spatial difference in terrestrial NEP has been reported from eddy-flux measurements, model outputs and atmospheric inversion products. In addition, the global average IAV of NEP was large relative to global annual mean NEP (Baldocchi et al., 2018). More importantly, the spatial variations of NEP and $IAV_{NEP}$ were typically underestimated by the compiled global dataset and the process-based global models (Jung et al., 2020; Fu et al., 2019)."

**Comment 4A:** *The IAV_NEP and beta for shrublands and savannas are among the smallest compared to other PFTs (Figure 3). Is it at odds with previous global studies that suggest semi-arid ecosystems contributed the most to global IAV_NEP? (Ahlström et al., 2015).*

Response: Thanks for this suggestion. As the reviewer has mentioned, there are very few semi-arid ecosystems (e.g. 2 shrublands and 5 savannas in the presented study) in the FLUXNET sites, while they represent a large portion of land at the global scale and have been shown to substantially control the interannual variability of NEP. Therefore, we have added several sentences in *Discussion Section* (Lines 238-241) to illustrate this point:

"However, the relatively lower $\beta$ in shrublands and savannas should be interpreted cautiously. There are very few semi-arid ecosystems in the FLUXNET sites, while they represent a large portion of land at the global scale and have been shown to substantially control the interannual variability of NEP (Ahlström et al., 2015)."

**Technical comments:**
**Comment 5A:** *In the legend of Figure 1 please indicate the source of NEP data.*

Response: This section has been removed.

**Comment 6A:** *L74. Do you mean the "relative differences" between photosynthesis and respiration or between their covariances?*

Response: Thanks, we have rephrased this sentence as "Because photosynthesis and respiration are strongly correlated over space (Baldocchi et al., 2015; Biederman et al., 2016), their relative difference could determine the spatial variation of NEP."

**Comment 7A:** *L100. Rephrase. "to address the local indicators"?*

Response: Thanks, we have rephrased this sentence as "In this study, we decomposed annual NEP into U and R, and explored the local indicators for spatially varying NEP."

**Comment 8A:** *L102. Reference for FLUXNET2015 is Pastorello et al., 2017.*

Response: Thanks. This sentence has been revised.

**Comment 9A:** *L84 -86. Generally, I feel there is a need to clarify why there is a need to find a local indicator (which is also a new phrase)? Does it help in the attribution of spatial variation of NEP and IAV_NEP to different processes, or does it provide an independent constrain on NEP and IAV_NEP?*

Response: Thanks for this valuable suggestion. The suggestion proposed by the reviewer inspires us to reorganize the importance of our work. We have added several sentences in the *Introduction Section* (Lines 81-86) to state the necessary of exploring the local indicators:

"However, despite the previous efforts in a predictive understanding of the land-atmospheric C exchanges, the multi-model spread has not reduced over time (Arora et al., 2019). Therefore, it is imperative to explore the potential indicators for the spatially varying NEP, which could help attribute the spatial variation of NEP and $IAV_{NEP}$ into different processes and provide valuable constraints for the global C cycle."

**Comment 10A:** *L135. I understand the scale-mismatch between model and eddy-covariance sites is difficult to address, but is it possible that muted spatial variation of NEP and IAV_NEP from gridded products is partly related to the scale mismatch?*

Response: Thanks for this suggestion.

First, considering the scale mismatch between FLUXNET sites and the gridded products, we have removed the direct comparison of the spatial variation of mean annual NEP and $IAV_{NEP}$ from different sources in *Section 3.3*. Instead, we mainly emphasize the important role of local indicators in indicating the spatially varying NEP.

Second, we have run the same analysis at the global scale based on the Jena Inversion product, the FLUXCOM product and the outputs of CLM4.5 model (Figure 1A). The results have strengthened our major conclusion that the spatial variation of mean annual NEP can be indicated by $ln$(U/R), while the spatial distribution of $IAV_{NEP}$ is well indicated by the slope (i.e., $\beta$) of the demonstrated logarithmic correlation. We have added the results of these new analyses into the *Results Section* (Lines 81-86) as Figure 6. The major revisions in the *Results Section 3.3* are cited as below:

"However, the spatial variations of NEP and $IAV_{NEP}$ were associated with the spatial resolution of the product (Marcolla et al., 2017). At the global scale, the spatial variation of mean annual NEP can be also well indicated by ln(U/R) (Fig. 6). The widely reported larger C uptake in FLUXCOM (Jung et al., 2020) resulted from its higher simulations for U/R. In addition, the larger spatial variation of $IAV_{NEP}$ in CLM4.5 could be inferred from the indicator $\beta$."

[Figure]

Figure 4B. Representations of the spatially varying NEP and its local indicators in FLUXCOM product and the Community Land Model (CLM4.5) at the global scale. **a**, The variation of mean annual NEP and $IAV_{NEP}$ derives from Jena Inversion, FLUXCOM and CLM4.5. Variation in mean annual NEP: the spatial variation of mean annual NEPs; Variation in $IAV_{NEP}$: the spatial variation of standard deviation in $IAV_{NEP}$. **b**, Representations of the local indicators for NEP in Jena Inversion, FLUXCOM and CLM4.5.

**Comment 11A:** *L229. "difference" -> "variation".*

Response: Done as suggested.

Dear editor:

Thank you very much for handling our manuscript. We really appreciate the reviewer for the invaluable suggestions and comments on our manuscript. Below, we address all the comments from reviewer #2 point-by-point. The comments are italicized and our response follow in blue, and we hope we could address the concerns from reviewer.

**Reply to Reviewer #2**

**General comments:**
**Comment 1B:** Erqian Cui et al. studied the annual NEP and the inter-annual variability of NEP and intended to provide local indicators to better understand their spatial patterns at the FLUXNET site level. I find this study relevant as it is important to have a better understanding of the factors controlling the spatial and inter-annual variability of NEP. However, I have some concerns about some aspects of the method and how the results are presented (see More specific comments section). In addition, there are some results presented in this study that do not provide ay significant new information compared to the available literature (e.g. spatial patterns of annual NEP and IAV of NEP at the global scale). Plus, most of the analysis is done at FLUXNET site level, therefore I do not really the point of using the FLUXCOM and CLM4.5 for the presented study. In short, although I find the presented study suitable for the scope of Biogeosciences, the manuscript is still in its early stage to be accepted as it is, therefore I suggest to make major revisions before potential acceptance.

Response: Thank you for the valuable suggestions. Based the reviewer's comment, we have made a substantial revision on both of the *Method* and *Results* sections.

First, we have deleted Figure 1 from *Results*, and moved the related contents to the *Introduction Section* as the background of our study.

Second, we have showed the major findings with FLUXNET observations and the atmospheric inversion product (i.e. the new results in Figure 1B). Then as suggested by the reviewer, we have benchmarked the simulations from the compiled global product and the process-based global model both at the global scale and at the FLUXNET site level (i.e. the new results in Figure 4B).

**Specific comments:**

**Comment 2B:** L. 3-4 The title is very confusing and does not really reflect the findings of the analysis. Please try to rephrase the title so that it matches the message the analysis is trying to convey.

Response: Thanks, we have revised the title as "Spatial variations in terrestrial net ecosystem productivity and its local indicators".

**Comment 3B:** L. 38 "machine-learning-derived database." This concept seems odd and confusing. What about something like "based on a compiled global dataset and a machine learning method". The use "'machine-learning-derived database' is also not entirely true because, as far as I understood, only the FLUXCOM dataset is based on machine learning approaches. FLUXNET in-situ data and the CLM4.5 product are not using any machine-learning methods.

Response: Thanks for pointing out this issue. We have rephrased the relevant statement as "based on daily NEP observations from FLUXNET sites and the atmospheric inversion product" in this version (Line 38).

**Comment 4B:** L. 65 "is related to the strength of carbon sink". It can also relate to the strength of the carbon source. Consider rephrasing to be more generic.

Response: Done. We have rephrased this sentence as "is related to the strength of carbon exchange" (Line 60).

**Comment 5B:** L. 68 Not convinced by the use of 'asynchronously' all over the manuscript, particularly because the results presented in the manuscript do not provide evidence that the spatial patterns of annual NEP or IAV_NEP are not simultaneous or concurrent in time.

Response: Done. We have deleted the word "asynchronously" all over the manuscript and replaced it with "variation".

**Comment 6B:** L. 76-77 'environmental fluctuations among years'. Musavi et al., 2017 attributed the year-to-year variation to species richness and stand age. In the same line, Besnard et al. 2018 attributed most of the annual NEP variation to forest age.

Response: Thanks. We have revised this sentence as "Many previous analyses have attributed the $IAV_{NEP}$ at the site level to the different sensitivities of ecosystem photosynthesis and respiration to environmental drivers (Gilmanov et al., 2005; Reichstein et al., 2005) and biotic controls (Besnard et al., 2018; Musavi et al., 2017)." (Lines 74-76).

**Comment 7B:** L. 82-84 Can this sentence be merged with the 1st sentence of the paragraph (L.71-72)? They seem quite redundant.

Response: In the former version, the first sentence illustrated the decomposition of NEP as the difference between photosynthesis and respiration, while the last sentence lead to the decomposition of NEP directly into $CO_2$ uptake flux and $CO_2$ release flux. To make these points clearer, we have rephrased this sentence on Lines 86-91 as:

"Alternatively, the annual NEP of a given ecosystem can be also directly decomposed into $CO_2$ uptake flux and $CO_2$ release flux (Gray et al., 2014), which are more direct components for NEP (Fu et al., 2019). Many studies have reported that the vegetation $CO_2$ uptake during the growing season and the non-growing season soil respiration are tightly correlated (Luo et al., 2014; Zhao et al., 2016). It is still unclear how the ecosystem $CO_2$ uptake and release fluxes would control the spatially varying NEP."

**Comment 8B:** L. 84-86 The last sentence of this paragraph seems a bit out of the context of the whole paragraph. Consider improving the transition between the last sentence of the paragraph and the entire paragraph.

Response: Done. We have rephrased this section and strengthened our points by adding the following sentences (Lines 81-91):

"However, despite the previous efforts in a predictive understanding of the land-atmospheric C exchanges, the multi-model spread has not changed over time (Arora et al., 2019). Therefore, it is imperative to explore the potential indicators for the spatially varying NEP, which could help attribute the spatial variation of NEP and $IAV_{NEP}$ into different processes and provide valuable constraints for the global C cycle. Alternatively, the annual NEP of a given ecosystem can be also directly decomposed into $CO_2$ uptake flux and $CO_2$ release flux (Gray et al., 2014), which are more direct components for NEP (Fu et al., 2019). Many studies have reported that the vegetation $CO_2$ uptake during the growing season and the non-growing season soil respiration are tightly correlated (Luo et al., 2014; Zhao et al., 2016). It is still unclear how the ecosystem $CO_2$ uptake and release fluxes would control the spatially varying NEP."

**Comment 9B:** L. 85 "could be integrated into some simple indicators". I would use the term 'decompose' instead of 'integrated'. After all, the authors want to decompose the contribution of a series of carbon uptake and carbon release metrics to annual NEP and IAV_NEP.

Response: Done as suggested.

**Comment 10B:** L. 98-99 Not sure that FLUXCOM products are the best to assess IAV_NEP. Please check Jung et al. 2020 to understand the issues of such products when looking at IAV_NEP. Why not using NEE derived from atmospheric inversions though (e.g. JenaCarboScope (Rödenbeck et al., 2018), CAMSv17r1 (Chevallier et al., 2005, 2019) and CarbonTracker-EU (Peters et al., 2010)). At least, we know that this data capture some processes that contribute to IAV_NEP, which are not being captured with eddy-covariance data (e.g. fire, $CO_2$ fertilization).

Response: The authors really appreciate the reviewer for this great suggestion. We have verified the relationship derived from FLUXNET sites with the Jena CarboScope $CO_2$ Inversion, and find that the relationship between annual NEP and $\frac{U}{R}$ is robust in most global grid cells. We have added these new analyses in the *Results Section* (Lines 193-198) and Figure 2 (i.e., the following Fig. 1B) to strengthen our findings:

"In addition, the relationship between NEP and $\frac{U}{R}$ was also verified by the atmospheric inversion product (i.e., Jena CarboScope Inversion). The control of $\frac{U}{R}$ on annual NEP was robust in most global grid cells (i.e. $0.6 < R^2 < 1$). The explanation of $\frac{U}{R}$ was higher in 80% of the regions, but lower in North American (Fig. 2). These two datasets both showed that the indicator $\frac{U}{R}$ could successfully capture the variability in annual NEP."

[Figure]

Figure 1B. Relationship between annual NEP and $\frac{U}{R}$ for Jena Inversion product (of the form $NEP = \beta \cdot \ln(\frac{U}{R})$). The black box indicates the location of the sample.

**Comment 11B:** L. 122-129 It might be relevant to specify that you use the FLUXCOM RS-meteo products for which the inter-annual variability is only driven by climatic conditions as they used the mean seasonal cycle of remote sensing products. This basically means that there is no inter-annual variability directly related to the state of vegetation.

Response: Done. We have rephrased the description of FLUXCOM product by adding the following sentences in *Method Section* (Lines 141-143):

"It should be noted that the inter-annual variability of FLUXCOM product is only driven by climatic conditions, the effects of land use and land cover change are not represented."

**Comment 12B:** L. 124 why only using the CRUNCEPv6 product. In my understanding, FLUXCOM uses more than one meteorological forcing as well as different machine-learning methods. Using all the FLUXCOM RS-meteo products could additionally provide uncertainty estimates for the presented indicators.

Response: Thanks for this comment. We used the CRUNCEPv6 product mainly due to two reasons. First, the simulations from CLM4.5 and Jena Inversion in this study are both driven by CRUNECP meteorological forcing. Therefore, in order to reduce the uncertainty caused by meteorological forcing, we would prefer to choose the CRUNCEPv6 product. Second, we have averaged all the FLUXCOM CRUNCEPv6 products with different machine-learning methods to avoid the uncertainty caused by machine-learning methods. To illustrate our consideration clearer, we have detailed the selection of the product in *Method Section* (Lines 138-141):

"To be consistent with the meteorological forcing of Jena Inversion product and the CLM4.5 model, we used the FLUXCOM CRUNCEPv6 products. In addition, in order to reduce the uncertainty caused by machine-learning methods, we averaged all the FLUXCOM CRUNCEPv6 products with different machine-learning methods."

**Comment 13B:** L. 122-136 If one of the aims is to compare FLUXCOM and CLM4.5, I would suggest comparing the two products during the same time period (i.e. 1990-2010).

Response: Thanks for this suggestion. We have adjusted the time period of all the global products to 1985-2010.

**Comment 14B:** L. 133 'match the available FLUXCOM dataset.' Spatially or temporally? As far as I know, the FLUXCOM products have a spatial resolution of either 0.5 or 0.0833 degrees (http://www.fluxcom.org/CF-Products/).

Response: Thanks. We have adjusted the global products to the same time period (1985-2010) and specified their spatial resolution in the *Method Section*.

**Comment 15B:** L. 140 equation 1: So U is conceptually GPP and R ecosystem respiration, right? I would be curious to see how GPP compared to U when U is computed as in equation 4 for a sanity check. Are they the same? In principle yes, right? Same for ER and R.

Response: Sorry for the confusion. We have drawn a concept figure to show our method to decompose the NEP in our study (Fig. 2B). The annual NEP is determined by vegetation photosynthesis and ecosystem respiration, but here we decompose the annual NEP into its more direct components: $CO_2$ uptake flux and $CO_2$ release flux. To describe the decomposition process more clearly, we have modified the decomposition process of NEP in *Method Section*.

[Figure]

Figure 2B. Conceptual figure for the decomposition of annual NEP in this study. The example shows daily observations from BE-Bra site.

**Comment 16B:** L. 143 I am not sure if this equation is written correctly. Assuming that U is supposed to be expressed in gC m-2 d-1, the way the equation is written suggests that the U would be expressed in gC m-2 (assuming that CUP is a length expressed in the number of days), which is then inconsistent with equation 4. Or did I misunderstand how CUP is calculated?

Response: Thanks for reminding the confusion of the units. In this study, $U$ is expressed in gC $m^{-2}$ $yr^{-1}$ and calculated from the mean daily $CO_2$ uptake ($\overline{U}$, gC $m^{-2}$ $d^{-1}$) over the carbon uptake period ($CUP$, d $yr^{-1}$). In fact,, the equations (2)-(3) and (4)-(5) are mathematically equivalent. Based on the suggestions from the reviewer, and in order to avoid using the ambiguous units, we have removed the original equations (4) and (5).

**Comment 17B:** L. 144 The same applies to this equation.

Response: Thanks, we have deleted equations (4) and (5).

**Comment 18B:** L. 148-149 I think these equations are correct and good enough to explain how U and R are calculated, therefore I would discard equation (2) and (3) to avoid confusion. Again, U and R derived from equations 2 and 3 do not seem to match how U and R are calculated from eq 4 and 5.

Response: Done. We have discard equations (2) and (3) to avoid confusion

**Comment 19B:** L. 150-153 "Because many studies have [..] are tightly correlated" I would move this sentence to the introduction. I am also not sure that this is enough to justify the need to look at the relationship between annual NEP and the ratio U/R.

Response: Thanks for this suggestion. We have removed these sentences to the *Introduction Section* and added several sentences to state the motivation to explore the relationship between annual NEP and its components *U* and *R* (Lines 81-91):

"However, despite the previous efforts in a predictive understanding of the land-atmospheric C exchanges, the multi-model spread has not changed over time (Arora et al., 2019). Therefore, it is imperative to explore the potential indicators for the spatially varying NEP, which could help attribute the spatial variation of NEP and $IAV_{NEP}$ into different processes and provide valuable constraints for the global C cycle. Alternatively, the annual NEP of a given ecosystem can be also directly decomposed into $CO_2$ uptake flux and $CO_2$ release flux (Gray et al., 2014), which are more direct components for NEP (Fu et al., 2019). Many studies have reported that the vegetation $CO_2$ uptake during the growing season and the non-growing season soil respiration are tightly correlated (Luo et al., 2014; Zhao et al., 2016). It is still unclear how the ecosystem $CO_2$ uptake and release fluxes would control the spatially varying NEP."

**Comment 20B:** L. 160 This equation is correct if one assumes that equations 2 and 3 correct, and if I understood correctly their formulation, equations 2 and 3 are not (see comment above). Therefore, I do not believe that the ratio U/R can be partitioned as presented in equation 7. It seems that part of the paper is based on assuming that equations 2 and 3 are correct, therefore I have concerned related to the analysis relying on equations 2 and 3.

Response: Thanks for this comment. To be consistent with the equation (7), we have deleted the equations (4) and (5) and kept the equations (2) and (3) as the final decomposition approaches.

**Comment 21B:** L. 171 I think the analysis presented in section 4 is not correct for the issues I have raised related to equations 2 and 3 at least the way equation 8 is expressed. One could express U/R = f(U/R, CUP/CUR) though and run the variable importance analysis. Why not just do the variable importance analysis as NEP = f(U/R, CUP/CUR)? I find it cleaner although it might be a bit circular and spurious as U and R are derived from NEP.

Response: Thanks for this valuable suggestion. In the revised version, we have directly tested the effect of these two ratios on the spatial variation in NEP (Figure 3B). These new results have been added in the *Results* as Figure 4. The major revisions in *Method Section* and *Results Section* are as below.

In the *Method* Section, please find the added sentences on Lines 175-179 as*:*

"We further quantified the relative contributions of $\frac{\bar{U}}{\bar{R}}$ and $\frac{CUP}{CRP}$ in driving the spatial variations in NEP:

$$\text{NEP} = \int (\frac{\bar{U}}{\bar{R}}, \ \frac{CUP}{CRP}) \qquad\qquad (6)$$

We used a relative importance analysis method to quantify the relative contributions of each ratio to the spatial variations in NEP."

In the *Results* Section, the added sentences could be found on Lines 206-210 as:

"The decomposition of indicator $\frac{U}{R}$ into $\frac{\bar{U}}{\bar{R}}$ and $\frac{CUP}{CRP}$ allowed us to quantify the relative importance of these two ratios in driving NEP variability. The linear regression and relative importance analysis showed a more important role of $\frac{CUP}{CRP}$ (58%) than $\frac{\bar{U}}{\bar{R}}$ (42%) in explaining the cross-site variation of NEP (Fig. 4). Therefore, the spatial distribution of mean annual NEP was mostly driven by the phenological rather than physiological changes."

[Figure]

Figure 3B. The relative contributions of the local indicators in explaining the spatial patterns of mean annual NEP. a, The linear regression between mean annual NEP with $\frac{CUP}{CRP}$ ($R^2 = 0.33$, $P < 0.01$) and $\frac{\bar{U}}{\bar{R}}$ ($R^2 = 0.25$, $P < 0.01$) across sites. b, The relative contributions of each indicator to the spatial variation of NEP. The number of site-years at each site is indicated with the size of the point.

**Comment 22B:** L. 186 I do not find this section relevant in the context of the study. Besides, most of the presented results are already well documented in the literature (e.g. Jung at al. 2020).

Response: Thanks for this suggestion. We have deleted this section from *Results*, and moved the related content to the *Introduction Section* (Lines 65-69):

> "Large spatial difference in terrestrial NEP has been reported from eddy-flux measurements, model outputs and atmospheric inversion products. In addition, the global average IAV of NEP was large relative to global annual mean NEP (Baldocchi et al., 2018). More importantly, the spatial variations of NEP and $IAV_{NEP}$ were typically underestimated by the compiled global dataset and the process-based global models (Jung et al., 2020; Fu et al., 2019)."

**Comment 23B:** L. 188 Be aware that the 'large carbon sinks' are very likely related to an artifact in the eddy-covariance datasets due to advection and storage issues. It might be relevant to discuss eddy-covariance data quality issues.

Response: Thanks for this suggestion. Because this section has been removed in this revised version, so we didn't further discuss the eddy-covariance data quality issues.

**Comment 24B:** L. 204 Would that make sense to discard the sites for which the logarithmic function does not provide a correlation >0.9 for robustness?

Response: Thanks, we have rephrased this sentence (Lines 190-192) as "The logarithmic correlations between annual NEP and $\frac{U}{R}$ were significant at all sites (Fig. 1a; Fig. S2), and ~90% of $R^2$ falling within a range from 0.7 to 1 (Fig. 1c)."

**Comment 25B:** L. 207-208 "This finding suggests that the mean annual ratio ln(U/R) is a good indicator for NEP and its spatial variation." Isn't it expected? I mean U and R are derived from NEP so you might expect that their ratio explains the annual variability of NEP, right?

Response: Thanks. We have rephrased the related sentences to make the statements clearer: (1) *Results Section 3.1*: "These two datasets both showed that the indicator $\frac{U}{R}$ could successfully capture the variability in annual NEP." (2) *Results Section 3.2*: "This finding suggested that the mean annual ratio $\ln\left(\frac{U}{R}\right)$ is a good indicator for cross-site variation in NEP."

**Comment 26B:** L. 218 Again, is this analysis being done on the extracted time series for each Fluxnet sites or globally? If the former, I do not really see the point of included results based on FLUXCOM or CLM4.5 for the purpose of the study. It would be interesting to run this analysis both at the global scale and at the Fluxnet level.

Response: Yes, the previous analysis in Figure 5 was based on the extracted time series for FLUXNET sites. We agree with the reviewer that it would be interesting to also run the analysis at the global scale. In this revised version, we have run the same analysis at the global scale based on Jena Inversion product, FLUXCOM product and CLM4.5 model (Figure 4B). The results have strengthened our major conclusion that the spatial variation of mean annual NEP can be indicated by ln(U/R), while the spatial distribution of $IAV_{NEP}$ is well indicated by the slope (i.e., β) of the demonstrated logarithmic correlation. We have added these new analyses in *Results Section* (Lines 219-225) as Figure 6. The major revisions in *Results Section* are as below:

> "In addition, the spatial variations of NEP and $IAV_{NEP}$ were associated with the spatial resolution of the product (Marcolla et al., 2017). Considering the scale mismatch between FLUXNET sites and the gridded product, we run the same analysis at the global scale based on Jena Inversion product. At the global scale, the spatial variation of mean annual NEP can be also well indicated by ln (U/R) (Fig. 6). The larger C uptake in FLUXCOM resulted from its higher simulations for ln(U/R). Furthermore, the larger spatial variation of $IAV_{NEP}$ in CLM4.5 could be inferred from the indicator $\beta$."

[Figure]

Figure 4B. Representations of the spatially varying NEP and its local indicators in FLUXCOM product and the Community Land Model (CLM4.5) at the global scale. **a,** The variation of mean annual NEP and $IAV_{NEP}$ derives from Jena Inversion, FLUXCOM and CLM4.5. Variation in mean annual NEP: the spatial variation of mean annual NEPs; Variation in $IAV_{NEP}$: the spatial variation of standard deviation in $IAV_{NEP}$. **b,** Representations of the local indicators for NEP in Jena Inversion, FLUXCOM and CLM4.5.

**Comment 27B:** L. 219 I do not think that one can directly compare the results from FLUXNET data and the two global products (i.e. FLUXCOM and CLM4.5) simply because of the strong bias in representativeness in the FLUXNET datasets. For instance, there are very few semi-arid ecosystems (e.g. 2 shrublands and 5 savannas in the presented study) in the FLUXNET dataset, while they represent a large portion of land at the global scale and have been shown to substantially control the interannual variability of NEP (Ahlström et al., 2015). Or do you extract FLUXCOM and CLM4.5 time series for each FLUXNET site location? If so, it is anyway not a fair comparison due to spatial mismatch as the footprint of a tower is definitely lower than 1 degree (CLM4.5) or 0.5 degree (FLUXCOM) spatial resolution. As previously mentioned, I would rather run this analysis globally and not only at FLUXNET sites to have a real added value by using global products such as FLUXCOM and CLM4.5.

Response: Thanks for pointing out the issue of scale mismatch. Considering the scale mismatch between FLUXNET sites and the gridded products, we have removed the direct comparison of the spatial variation of mean annual NEP and $IAV_{NEP}$ from different sources in *Section 3.3*. Instead, we mainly emphasized the important role of local indicators in indicating the spatially varying NEP.

Also, as suggested by the reviewer, we have done the same analysis both at the global scale and at the FLUXNET site level. The results from FLUXNET sites are used to benchmark the simulations of FLUXCOM product and CLM4.5 model at the FLUXNET site level, and the results from Jena Inversion product are used to evaluate the simulations of FLUXCOM product and CLM4.5 model at the global scale. As shown in Figure 4B, the analyses at the global scale and at the FLUXNET site level both support our major conclusion that the spatial variation of mean annual NEP can be indicated by $\ln(U/R)$, while the spatial distribution of $IAV_{NEP}$ is well indicated by the slope (i.e., $\beta$) of the demonstrated logarithmic correlation.

**Technical corrections:**
**Comment 28B:**

L.57 'However' does not sound appropriate. Maybe 'furthermore' or 'in addition'.
Response: Done as suggested.

L. 62 'dramatic'. Try to avoid emotional semantic in a scientific paper. Maybe 'substantial' instead?
Response: Done as suggested.

L. 77. replace Musavi, 2017 by Musavi et al., 2017
Response: Done as suggested.

L. 104 'database' Replace database by product.
Response: Done as suggested.

L. 119-121 Stand age information is mentioned here but is they even being used further in the analysis? If not, please remove it.
Response: Done. We have removed it.

L. 154-155 'Then we found that annual NEP [...] (Figure S2).' To me, this already belongs to the results section.
Response: Thanks, we have removed this sentence to the *Results Section*.

L. 154 'the ratio U/R'. It might be relevant for the reader to see a sentence explaining the meaning of the ratio U/R. This explanation in L. 162-163 comes a bit too late.
Response: We have added the meaning of ratio U/R as "we further tested the relationship between annual NEP and the ratio of U/R. Ecologically, the ratio of U/R reflects the relative strength of the ecosystem $CO_2$ uptake." on line 158-159.

L. 151-152 'the non-growing soil respiration' Is that what you mean here? Maybe rephrase.
Response: We have rephrased it as "the non-growing season soil respiration".

L. 208 I would not say 'was well explained' but rather that the correlation was moderate (i.e. $0.3 > r > 0.7$)
Response: We have rephrased it as "was moderately explained".

L. 347 In Fig. 1, it is not clear to me what products are we looking at. FLUXCOM, CLM 4.5 or both? It seems to be FLUXCOM (L. 99) but please specify in the figure's caption.
Response: As suggested by Comment 22B, we have deleted Figure 1 and the related results.

**References:**

[revised manuscript text omitted]

---

## Author Response (AR2)

**Response to Editorial Review**

Dear editor:

Thank you very much for handling our manuscript. We really appreciate the insightful comments and suggestions from you and the reviewer. Below, we address the comments point-by-point, and the comments are italicized and our response follow in blue.

Major points

Comment 1: *As has been raised before, I think some discussion is necessary as to whether it is not to be expected that there is such a correlation as U and R are derived from NEP.*

Response: Thanks for this suggestion. As shown in Figure 2, this method was applied on the atmospheric inversion product. However, some ecosystems seem to defy such correlation. Therefore, we think the robustness in relationship between annual NEP and U/R depend on the stability of carbon uptake sensitivity for an ecosystem. We have added discussions as (Lines 297-303): "In this study, the atmospheric inversion product shows low correlation between NEP and $\ln\left(\frac{U}{R}\right)$ in some boreal ecosystems, which might due to that the atmospheric inversion product is failed to capture the carbon uptake sensitivity in these boreal ecosystems or these boreal ecosystems are experiencing serious disturbances. Therefore, the robustness in relationship between annual NEP and $\ln\left(\frac{U}{R}\right)$ depends on the temporal stability of carbon uptake sensitivity for an ecosystem. In addition, the spatial variation in β reveals the differences of carbon uptake sensitivity across ecosystems".

Comment 2: *Please provide a derivation of Eq 4. This does not logically follow from Eq. 1*

Response: Thanks for this suggestion. To avoid logic misleading, we have deleted Eq.1. In addition, we have added some sentences to illustrate the motivation of testing the relationship between annual NEP and the ratio *U/R* (Lines 159-167): "Many studies have reported that the vegetation net $CO_2$ uptake during the growing season and the non-growing season soil net $CO_2$ release are tightly correlated (Luo et al., 2014; Zhao et al., 2016). Therefore, we further tested the relationship between annual NEP and $\frac{U}{R}$ (i.e., $NEP \propto \frac{U}{R}$), which reflects the seasonal carbon uptake-release ratio. Consequently, NEP in any given ecosystem can be expressed as (Fig. S2):

$$NEP = \beta \cdot \ln\left(\frac{U}{R}\right) \tag{3}$$

where the parameter $\beta$ represents the slope of the linear relationship of $NEP \propto \ln\left(\frac{U}{R}\right)$, indicating the site-level carbon uptake sensitivity".

Minor points

*L41 rephrase to "large-scale estimates from an atmospheric inversion product"*

Response: Thanks, done as suggested.

*L44: linearly related to ln (X) is equivalent to logarithmically related to X?*

Response: Thanks, and we have rephrased this sentence as "NEP could be logarithmically indicated by U/R".

*L45: beta has not been defined (the slope of what?). Explain "well indicated"*

Response: Thanks, and we have rephrased this sentence as "while the spatial distribution of $IAV_{NEP}$ was associated with the slope (i.e., $\beta$) of the logarithmic correlation between annual NEP and $U/R$".

*L49: gridded products of what?*

Response: Thanks, and we have revised it as "gridded NEP products".

*L77: unclear what "the compiled" refers to here*

Response: Thanks, and we have revised it as "the global flux tower-based product".

*L82: Isn't this a contradiction? If they are strongly correlated in space, then how would they determine the spatial variation in NEP?*

Response: We have rephrased this sentence as (Lines 72-75): "The NEP in terrestrial ecosystems is determined by two components, including vegetation photosynthesis and ecosystem respiration (Reichstein et al., 2005), and their relative difference could determine the spatial variation of NEP (Baldocchi et al., 2015; Biederman et al., 2016)".

*L107: here and in the following: the total NET uptake. Also this isn't entirely correct because as you note later also the strength of the source/sink within each period can vary.*

Response: Thanks for this suggestion. We have rephrased the description as "the total net $CO_2$ uptake flux ($U$)" and "the total net $CO_2$ release flux ($R$)" in the whole paper.

*L110: I don't understand the use of the word "innovatively attributed" here. Please clarify*

Response: We have rephrased it as "The variations of NEP thus could be attributed to these decomposed components".

*L146: Add "to infer the net CO2 exchanges between land, ocean and atmosphere at large scales*

Response: Done as suggested.

*L165ff: Is this correct? Are the effects of land cover changes not implicitly included by the use of satelite derived fAPAR?*

Response: Thanks. We have revised the description of FLUXCOM product as ( Lines 140-144): "It should be noted that the inter-annual variability of FLUXCOM product is driven by meteorological measurements and satellite data, which partially includes information on vegetation state and other land surface properties".

*L168: Please provide correct link to the data portal*

Response: Thanks. We have revised the link of FLUXCOM product.

*L190: A quantitative definition of CUP and CRP is missing here.*

Response: Thanks. We have added the definition of CUP and CRP as "where *CUP* (d $yr^{-1}$) is the length of $CO_2$ uptake period and *CRP* (d $yr^{-1}$) is the length of $CO_2$ release period".

*L254: Verified is the wrong word here*

Response: We have revised it as "confirmed".

*L272: use "across-site variation" instead to spatial change?*

Response: Done as suggested.

*L283: The use of "mostly" is inappropriate here, because CUP/CRP explains less than 60% of the variance.*

Response: We have rephrased it as "Therefore, the spatial distribution of mean annual NEP was more strongly driven by the phenological changes".

*L317: But what if most of this crop IAV is related to changes in the local crop from year to year and are therefore not representative of regional scale cropland IAV?*

Response: Sorry for the confusion. We have rephrased this sentence as (Lines 247-249): "The highest $\beta$ implies that the land covered by cropland with the largest $IAV_{NEP}$. Therefore, the reported rapid global expansion of cropland may enlarge the fluctuations in Land-atmosphere $CO_2$ exchange".

*L325/L326: These statements need to be adjusted to reflect that the difference in explanatory power is "only" 58 to 42%.*

Response: Thanks for this suggestion.

First, we have rephrased the subtitle as "Joint control of plant phenology and physiology on mean annual NEP".

Second, we have revised these sentences to emphasize the equal importance of plant phenology and physiology in driving the spatial difference of mean annual NEP as (Lines 255-257): "Here we demonstrated that the spatial difference of mean annual NEP was determined by both the phenology indicator $\frac{CUP}{CRP}$ (58%) and the physiological indicator $\frac{\overline{U}}{R}$ (42%). In addition, the lower contribution of the physiological indicator could partly be attributed to the convergence of $\frac{\overline{U}}{R}$ across FLUXNET sites (Fig. S4)".

**Response to comments from reviewer #1**

Comment 1: *The manuscript proposed to study the relationship between ln(U/R) and NEP. Since we know that NEP = U – R and ln(U/R) = ln(U) – ln(R), therefore it is expected to see a strong r2 between ln(U/R) and NEP (Figure 1-3). I am more curious about why some ecosystems (i.e. boreal ecosystems in Figure 2) seems to defy such correlation, and why the slope of this correlation (i.e. beta) changes spatially? Further discussions on these would be appreciated.*

Response: Thanks for this valuable suggestion.

For any year of each site, the indicator β was equivalent to the quotient between annual NEP and ln(U/R). Generally, the indicator β was convergent within-site and represents the site-level carbon uptake sensitivity. However, the indicator β would shift when an ecosystem experiences the serious disturbance, such as extreme heat waves and drought (Figure R1).

Therefore, the atmospheric inversion product presents low correlation between NEP and ln(U/R) in some ecosystems because of the following two reasons: 1) the atmospheric inversion product was failed to capture the carbon uptake sensitivity in these boreal ecosystems; 2) these boreal ecosystems were experiencing serious disturbance that affect their carbon sink stability. In addition, the spatial variation in β reveals the differences of carbon uptake sensitivity across ecosystems.

We have added these discussions in the revised manuscript (Lines 296-303): "Thus, a sudden shift of the *β*-value may be an important early-warning signal for the critical transition of carbon uptake sensitivity of an ecosystem. In this study, the atmospheric inversion product shows low correlation between NEP and ln(U/R) in some boreal ecosystems, which might due to that the atmospheric inversion product is failed to capture the carbon uptake sensitivity in these boreal ecosystems or these boreal ecosystems are experiencing serious disturbances. In addition, the spatial variation in β reveals the differences of carbon uptake sensitivity across ecosystems".

[Figure]

Figure R1. (a) The relationship between annual NEP and ln(U/R) at a specific site. (b, c) Shift of indicator β in some specific sites along soil moisture and temperature.

Comment 2: *For latter, the variation in beta is suggested to be related to IAV_NEP, which is regarded as an indicator of the carbon sink stability. Wouldn't IAV_NEP normalized by mean NEP make more sense here? I would be curious to see if there is a relationship between normalized IAV_NEP and beta, as sites with larger NEP seem more likely to have larger beta.*

Response: Thanks for this suggestion.

First, the site-level mean annual NEP includes both negative and positive values, and therefore the IAV_NEP was quantified as the standard deviation of annual NEP rather than the normalized value. This approach have been widely used in the previous studies (Baldocchi et al., 2018; Marcolla et al., 2017).

Second, as suggested by the reviewer, we have tested the relationship between mean annual NEP and β, and found low correlation between mean annual NEP and β (Figure R2).

Third, the IAV_NEP in this study represents both the intensity and amplitude of variation in terrestrial carbon sink. Therefore, we prefer to use standard deviation of annual NEP to represent its inter-annual variation.

[Figure]

Figure R2. The relationship between mean annual NEP and the indicator β.

Marcolla, B., Rödenbeck, C., & Cescatti, A. (2017). Patterns and controls of inter-annual variability in the terrestrial carbon budget. Biogeosciences, 14(16), 3815-3829.

Baldocchi, D., Chu, H., & Reichstein, M. (2018). Inter-annual variability of net and gross ecosystem carbon fluxes: A review. Agricultural and Forest Meteorology, 249, 520-533.

Comment 3: *Figure 3a presents the correlation between annual mean NEP and ln(U/R) across sites. I think there is a need to clarify the calculation of ln(U/R) here as this indicator changes year-to-year for each site (did you use the mean ln(U/R) of each site?).*

Response: Yes, Figure 3a shows the spatial correlation between annual mean NEP and mean ln(U/R) of each site. We have added this information in Lines 204-205 as "Across the 72 flux-tower sites, the across-site variation in mean annual NEP were significantly correlated to mean annual $\ln\left(\frac{U}{R}\right)$ of each site ($R^2 = 0.65$, $P < 0.01$) (Fig. 3a)".

Comment 4: *It would be helpful to give more details on Equation (6). I could not understand how to decompose ln(U/R) into the two components from what is presented here, but the method seems critical to Figure 4. Is equation (6) a multivariate linear function?*

Response: Thanks for this suggestion.

First, we have revised the expression of equation (6) as (Lines 161-167): "We further quantified the relative contributions of $\frac{\overline{U}}{\overline{R}}$ and $\frac{CUP}{CRP}$ in driving the spatial variations in NEP:

$$\text{NEP} = \beta \cdot [\ln\left(\frac{\overline{U}}{\overline{R}}\right) + \ln\left(\frac{CUP}{CRP}\right)] \tag{5}$$

For a specific ecosystem, the parameter $\beta$ was constant. Then, we used a relative importance analysis method to quantify the relative contributions of these two ratios to the spatial variations in NEP".

Second, the reviewer was right that we quantified the contributions of explanatory variables with a multiple linear regression model. The method was illustrated in Lines 186-189.

[revised manuscript text omitted]

---

## Author Response (AR3)

**Response to Editorial Comments**

Dear editor:

Thank you very much for handling our manuscript, and we really appreciate the insightful comments from you. Below, we address the comments point-by-point, and the comments are italicized and our response follow in blue.

**Minor comments:**

*L121: check english language: use combine instead of compiles from?*
*L168: remove, "Ecologically"*
*L180: define what "a specific ecosystem" means*
*L193ff: check grammar*
*L198: define "explanation". Is that explained variance?*
*L199: North America*
*L204: add "in this network".*
*L259: reword "which validates the discovery" to "which confirms / agrees with" or similar*

Response: Thanks, and we have made revisions according to the comments.

*L261: explain why the difference in U/R across ecosystem types indicates spatial convergence. The SD for individual ecosystem types is for some at least larger than for the total - this does not suggest that the variance across ecosystems is generally larger than within ecosystems.*

Response: We feel sorry for the confusion. We have removed the comparison between ecosystems and rephrased this sentence by emphasizing the convergent $\frac{\overline{U}}{R}$ across Fluxnet sites (Lines 255-258): "In this study, we found the $\frac{\overline{U}}{R}$ across the 72 sites is 2.71 ± 1.61, which confirms with the findings of Churkina *et al*. This spatial convergence of $\frac{\overline{U}}{R}$ at site level provides important constraints for global models that simulate large spatial variation in physiological processes (Peng et al., 2015; Xia et al., 2017).".

*L295: I am not convinced that the lack of a correlation for boreal ecosystems based on one atmospheric inversion is necessarily an indication of the inversions incapacity to reproduce observed carbon fluxes at regional scales. It is also possible, that the correlation does simply not exist (fluxnet is not well constrained for these regions. It is also unclear what "experiencing serious disturbance" refers to. Please be more specific.*

Response: Thanks for this suggestion. We have revised the explanation for the lower correlation in boreal ecosystems by considering the editorial suggestion (Lines 291-294): "In this study, the atmospheric inversion product shows low correlation between NEP and $\ln\left(\frac{U}{R}\right)$ in some boreal ecosystems, which might due to that the terrestrial NEP

is not well constrained for these regions or these boreal ecosystems are experiencing state transition.".

*Data availability statement. Add a statement referring to FLUXCOM and inversion data availability*

Response: Thanks, and we have added data availability for FLUXCOM and inversion product (Lines 315-319).